# Interference with ERK-dimerization at the nucleocytosolic interface targets pathological ERK1/2 signaling without cardiotoxic side-effects

Angela Tomasovic et al.[#]

Dysregulation of extracellular signal-regulated kinases (ERK1/2) is linked to several diseases including heart failure, genetic syndromes and cancer. Inhibition of ERK1/2, however, can cause severe cardiac side-effects, precluding its wide therapeutic application. ERK$^{T188}$-autophosphorylation was identified to cause pathological cardiac hypertrophy. Here we report that interference with ERK-dimerization, a prerequisite for ERK$^{T188}$-phosphorylation, minimizes cardiac hypertrophy without inducing cardiac adverse effects: an ERK-dimerization inhibitory peptide (EDI) prevents ERK$^{T188}$-phosphorylation, nuclear ERK1/2-signaling and cardiomyocyte hypertrophy, protecting from pressure-overload-induced heart failure in mice whilst preserving ERK1/2-activity and cytosolic survival signaling. We also examine this alternative ERK1/2-targeting strategy in cancer: indeed, ERK$^{T188}$-phosphorylation is strongly upregulated in cancer and EDI efficiently suppresses cancer cell proliferation without causing cardiotoxicity. This powerful cardio-safe strategy of interfering with ERK-dimerization thus combats pathological ERK1/2-signaling in heart and cancer, and may potentially expand therapeutic options for ERK1/2-related diseases, such as heart failure and genetic syndromes.

[#]A full list of authors and their affiliations appears at the end of the paper.

As the Raf1/MEK/ERK1/2 signaling cascade is causatively involved in frequent diseases such as heart failure and cancer, but also in rare developmental syndromes (so called RASopathies), it is an important therapeutic target[1]. Several pharmacological antibodies blocking the extracellular domain of the epidermal growth factor receptor tyrosine kinase or small molecules targeting kinases of this signaling cascade with differential selectivity are available. However, the wide therapeutic application of these compounds is limited by side effects, inducing the development of severe cardiomyopathy with decreased ejection fraction and the development of resistances to these compounds[2–9]. In the heart, the Raf/MEK/ERK1/2 cascade is involved in adverse remodeling, but is also responsible for cardiomyocyte survival and protection from stress-induced cardiomyocyte death. Inhibition of Raf/MEK/ERK1/2 signaling by above-mentioned inhibitors renders the heart more vulnerable to injury, but conversely, activation of the cascade can also be a trigger for adverse cardiac remodeling and subsequent cardiac dysfunction[10–12]. Thus, suppression of Raf/MEK/ERK1/2-mediated pro-hypertrophic and adverse remodeling processes and preservation of its pro-survival events are clearly warranted, but still remain an unmet clinical need.

Previously, we showed that autophosphorylation of ERK1/2 at threonine 188 in mouse ERK2 (resp. T208 in mouse ERK1; subsequently referred to as ERK$^{T188}$-phosphorylation) is central for nuclear ERK1/2 target activation and for inducing ERK1/2-mediated pathological cardiac hypertrophy in response to several pathological stimuli[10,13,14]. These include several Gq- and Gs-coupled G protein coupled receptors (GPCR) that activate ERK1/2 by the phosphorylation of the so-called highly conserved TEY motif (resp. T183/Y185 in mouse ERK2 or often referred to as T202/Y204 in human ERK1)[15–17], thereby triggering homo- or hetero-dimerization of ERK1/2[10] and subsequently facilitating intermolecular ERK autophosphorylation at threonine 188. Experimental data using a phosphorylation-deficient mutant (ERK2$^{T188A}$), that is dominant-negative for ERK1 and ERK2 autophosphorylation at threonine 188[10] suggested that interference with ERK$^{T188}$-phosphorylation may offer a possibility to selectively target pathological cardiac hypertrophy without affecting anti-apoptotic signaling of ERK1/2[10]. Interestingly, ERK2$^{T188A}$ only inhibited nuclear ERK1/2 target phosphorylation whilst cytosolic signaling remained unaffected.

Even though many mechanistic details of the Raf/MEK/ERK1/2 signaling cascade have been discovered, no therapeutic tools are currently available to differentially target this cascade in the nucleus and cytosol. In order to design a cardio-safe ERK1/2 inhibitor with new or extended indications, we here establish a molecular strategy that acts at the nucleocytosolic interface of ERK1/2 by interference with ERK dimerization. Using this peptide-based strategy, we show that (i) the ERK-ERK interaction is a prerequisite for ERK$^{T188}$-autophosphorylation which in turn is a central molecular event for nuclear ERK localization and signaling, (ii) ERK$^{T188}$-phosphorylation is also upregulated in human colon and lung cancer, and (iii) interference with nuclear ERK signaling is a cardio-safe targeting strategy to prevent pathological ERK1/2 signaling in the heart and in tumor cells. This strategy holds therapeutic potential for heart failure and possibly cancer therapy and other ERK1/2-related diseases.

## Results

### ERK2-Δ4 reduces cardiomyocyte hypertrophy but not survival.
As ERK dimerization is a prerequisite for pathological ERK$^{T188}$-phosphorylation[13], we set out to evaluate the ERK-ERK interface as a therapeutic target within the Raf/MEK/ERK1/2 signaling cascade to reduce maladaptive cardiac remodeling without causing apoptosis or cardiotoxic side-effects by ERK1/2 inhibition. We first analyzed the impact of monomeric ERK2 on cardiomyocyte hypertrophy: to this end, we used a non-dimerizing monomeric ERK2 mutant with a four amino acid deletion within the putative ERK-ERK interface, ERK2$^{Δ174-177}$ (ERK2-Δ4)[18]. ERK2-Δ4 is catalytically intact comparable to wild-type ERK2 (ERK2-wt) (pERK1/2[TEY]), but cannot undergo phosphorylation at threonine 188 as shown in response to phenylephrine stimulation (PE) in neonatal rat cardiomyocytes (NRCM) (Supplementary Fig. 1A, B; 10 min and 24 h). In line with the reduced extent of ERK$^{T188}$-phosphorylation, NRCM transduced with ERK2-Δ4 showed lower phosphorylation levels of the nuclear ERK1/2 target ETS (E 26)-like 1 transcription factor (ELK1)[19] as well as an attenuated hypertrophic response with respect to cell size and protein synthesis compared to ERK2-wt transduced cells (Fig. 1a and Supplementary Fig. 1A, C). Cardiomyocyte apoptosis, however, was comparable in ERK2-wt and ERK2-Δ4 transduced NRCM (Fig. 1b). In contrast, the MEK inhibitor PD98059 that inhibits ERK1/2 activation, abolishes – and this is different to ERK2-Δ4 – catalytic ERK1/2 activation and ERK$^{T188}$-phosphorylation and, subsequently, the hypertrophic response (Supplementary Fig. 1D, E) and increases cell death in response to H$_2$O$_2$ treatment (Supplementary Fig. 1F; also refer to ref.[10]).

Thus, monomeric ERK2 leads to an efficient attenuation of PE-induced cardiomyocyte hypertrophy without obvious adverse effects on cardiomyocyte survival.

### Expression of ERK2-Δ4 attenuates cardiac remodeling.
To assess the function and potential side-effects of monomeric ERK2 in the heart, we generated mice with cardiac-specific ERK2-Δ4 expression (ERK2-Δ4-tg; Supplementary Fig. 2A) and subjected ERK2-Δ4-tg and non-transgenic control mice (Wt) to transverse aortic constriction (TAC), a stimulus for chronic pressure overload resulting in cardiac remodeling[20,21]. Echocardiographic fractional shortening and ventricular diameters of ERK2-Δ4-tg were undistinguishable from Wt mice, and velocities of contraction and relaxation, even in response to dobutamine stress, were similar (Fig. 1c and Supplementary Table 1). The hypertrophic response to TAC of ERK2-Δ4-tg in comparison to Wt mice, however, was significantly attenuated, as monitored by heart weight-to-tibia length and heart weight-to-body weight ratios, echocardiographic left ventricular wall thickness and cardiomyocyte size assessed by histology (Supplementary Table 1 and Fig. 1d). Interstitial fibrosis and mRNA expression levels of brain natriuretic peptide (Nppb) and collagen (Col3a1) that are common markers for pathological remodeling[10,13] were also reduced in ERK2-Δ4-tg mice after TAC compared to Wt animals (Fig. 1e, Supplementary Fig. 2B, C and Supplementary Table 1). Cardiomyocyte death, however, was comparable in ERK2-Δ4-tg and Wt mice as shown by TUNEL and caspase-3 activity analyses (Fig. 1f and Supplementary Fig. 2D).

Of note, physiological growth of the heart that is needed to adapt the heart to altered work load as during exercise or pregnancy and that does not impair cardiac function and integrity[22], was not affected by ERK2-Δ4. Physiological growth of mouse hearts at least until the age of 9 months, or in response to exercise, i.e. a period of voluntary running in a wheel for three weeks, remained normal in ERK2-Δ4 overexpressing mice. In addition, ERK2-Δ4 expression did not affect cardiac function, degree of interstitial fibrosis or Nppb and Col3a1 transcription levels (Supplementary Tables 2, 3, Supplementary Fig. 2E–H). Also, survival of ERK2-Δ4-tg was comparable to non-transgenic Wt mice (Supplementary Fig. 2I). Of note, all experimental parameters assessed in Wt and ERK2-wt overexpressing mice have previously been shown not to differ, i.e. under basal

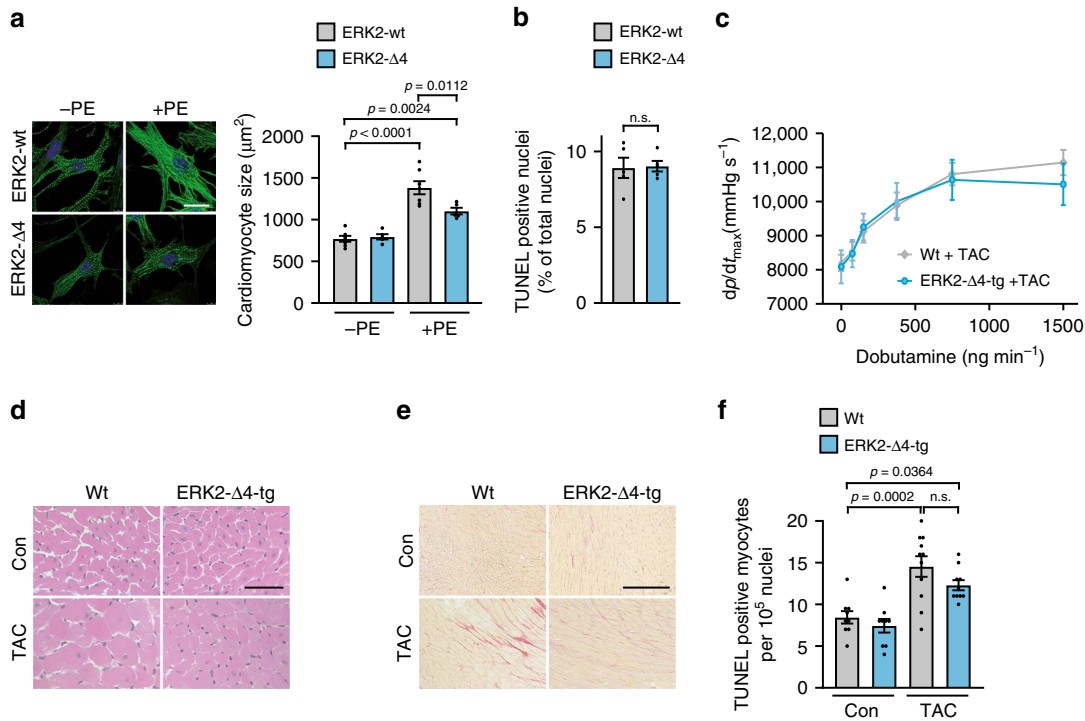

**Fig. 1 Monomeric ERK2$^{\Delta174-177}$ reduces cardiac hypertrophy but not cell survival. a, b** Analyses of cardiomyocyte size and apoptosis. Neonatal rat cardiomyocytes (NRCM) were transduced with Flag-tagged ERK2 wild-type (ERK2-wt) or the Flag-tagged monomeric ERK2 mutant, Flag-ERK2$^{\Delta174-177}$ (ERK2-Δ4), and treated with phenylephrine (PE, 4 μM, 24 h; **a**) or hydrogen peroxide (H$_2$O$_2$, 100 μM, 1 h; **b**) as indicated. **a** Representative phalloidin-stained NRCM (scale bar 20 μm) and analysis of cell size (ERK2-wt, $n = 7$; ERK2-Δ4, $n = 5$, 50–70 cells per group and experiment). **b** Analysis of TUNEL-positive NRCM ($n = 5$ and 500–1000 cells per group and experiment). **c–f** Analyses of cardiac function, cardiomyocyte size, interstitial fibrosis, and apoptosis of wild-type mice (Wt) and mice with cardiac-specific overexpression of ERK2$^{\Delta174-177}$ (ERK2-Δ4-tg). Mice were analyzed either untreated at the age of 8 weeks (Con) or 6 weeks after subjection to transverse aortic constriction (TAC). **c** Speed of left ventricular pressure rise ($dp/dt_{max}$) determined by left ventricular cardiac catheterization in response to dobutamine (Wt + TAC, $n = 13$; ERK2-Δ4-tg + TAC, $n = 8$ mice); **d, e** representative histological H&E (scale bar 50 μm, **d**) and Sirius Red (scale bar 200 μm, **e**)-stained sections of left ventricular myocardium, quantification in Supplementary Table 1; **f** analysis of TUNEL-positive cardiomyocyte nuclei (Con, $n = 9$; ERK2-Δ4-tg, $n = 9$; Wt + TAC, $n = 11$; ERK2-Δ4-tg + TAC, $n = 10$ mice per group. Error bars are mean ± s.e.m. For statistical analysis ordinary one-way ANOVA and Bonferroni as post hoc test (**a, c, f**) or unpaired two-sided Student's $t$-test (**b**) were applied. $n$ numbers in **a** and **b** represent biologically independent experiments. Source data are provided as a Source Data file.

conditions, after TAC or with regard to physiological hypertrophy, therefore, we used Wt littermates as controls within this study[10,13].

Hence, monomeric ERK2 expression has no adverse effects on cell survival, cardiac function, or remodeling and does not interfere with normal postnatal or exercise-induced cardiac growth even though it attenuates cardiomyocyte hypertrophy in response to pathophysiological hypertrophic stimuli such as phenylephrine or chronic pressure overload.

**pERK$^{T188}$ has strong impact on hypertrophic gene expression.** The introduction of an aspartic acid (D) in ERK2 mimics ERK$^{T188}$-autophosphorylation[13]. Interestingly, NRCM transduced with ERK2-Δ4 that additionally simulates a constitutive ERK$^{T188}$-phosphorylation (ERK2-Δ4D) showed – in contrast to ERK2-Δ4 – comparable hypertrophic responses to PE as NRCM transduced with ERK2-wt (Fig. 2a and Supplementary Fig. 3). Nuclear localization studies of ERK2-wt, ERK2-Δ4, and ERK2-Δ4D, N-terminally tagged with yellow fluorescent protein (YFP; YFP-ERK2-wt, YFP-ERK2-Δ4, and YFP-ERK2-Δ4D) in COS7 cells and in NRCM further revealed that ERK$^{T188}$-phosphorylation is essential for the nuclear ERK2 accumulation, but not sufficient (also refer to Supplementary Fig. 6H): PE stimulation led to a significant nuclear localization of YFP-ERK2-wt and ERK2-Δ4D but not of monomeric ERK2-Δ4 (Fig. 2b and

Supplementary Fig. 4A–C). These experiments identify ERK$^{T188}$-phosphorylation as the decisive trigger for nuclear ERK localization and cardiomyocyte hypertrophy and endorse ERK$^{T188}$-autophosphorylation as a potential target to specifically interfere with nuclear ERK signaling. To further evaluate the different outcomes of detrimental ERK-activating stimuli involving ERK$^{T188}$-phosphorlyation and more physiological ERK-activating stimuli, we used ERK2$^{T188D}$ + PE as a pathological stimulus involving ERK$^{T188}$-signaling and MEK1$^{SS218/222DD}$ (MEK-DD) and IGF as adaptive/physiological ERK1/2 stimuli[23,24]. MEK-DD is a constitutively active MEK1 mutant that can activate ERK1/2 independently from extracellular signals such as GPCR/G-protein activation. The insulin-like growth factor IGF has been associated with a physiological type of cardiac hypertrophy[25]. Expression levels or concentrations of ERK2$^{T188D}$ + PE, MEK-DD and IGF were adjusted for comparable effects on cardiomyocyte hypertrophy (Fig. 2c). Under these conditions, we evaluated YFP-ERK2 localization, ERK(T188) and ERK(TEY) phosphorylation, and expression of genes known to be involved in pro-hypertrophic signaling. While comparable pERK(TEY) levels were detected in response to all three hypertrophic stimuli, only ERK2$^{T188D}$ + PE resulted in increased ERK$^{T188}$-phosphorylation levels (Fig. 2d). In line with the induction of pERK$^{T188}$, YFP-ERK2-wt showed significant nuclear localization in cells treated with ERK2$^{T188D}$ + PE but not in the presence of

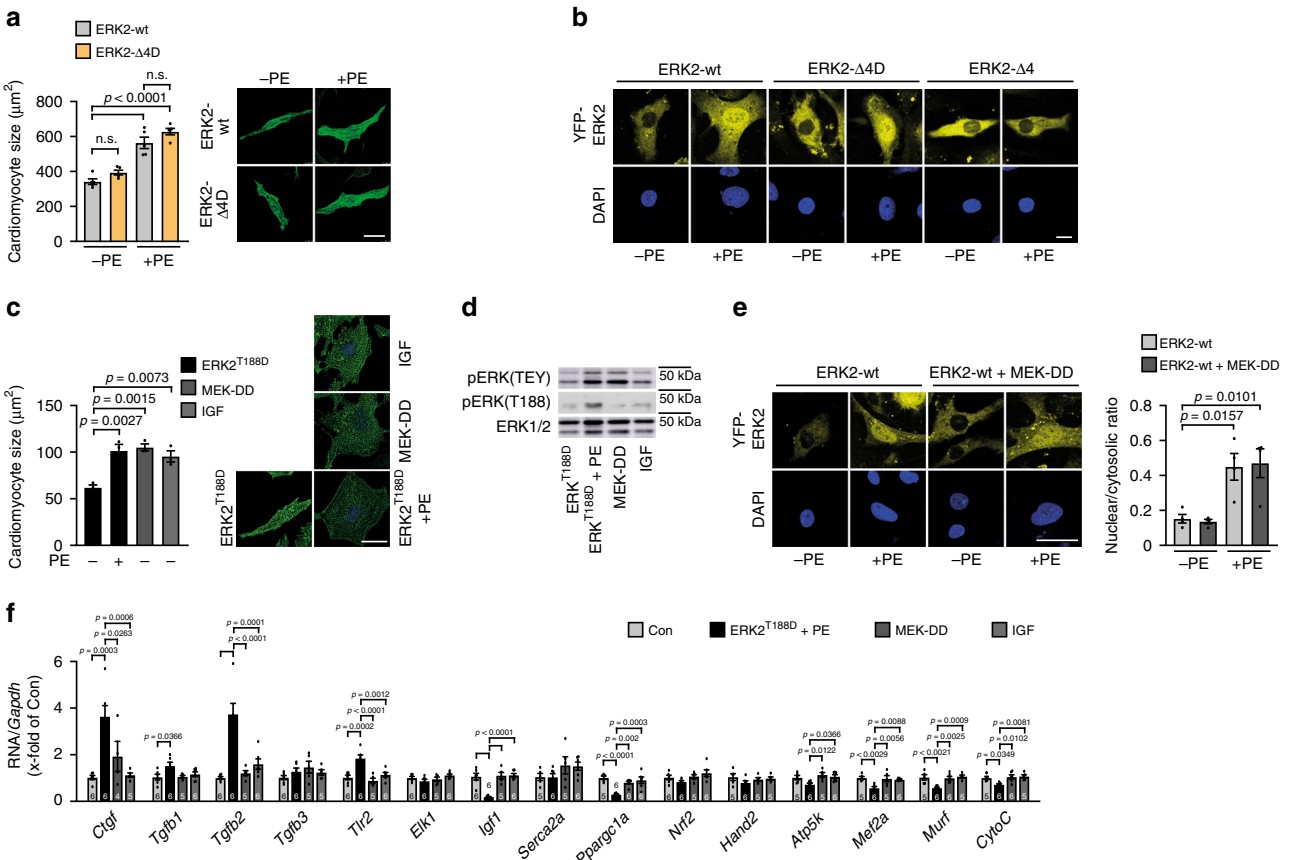

**Fig. 2 Nuclear ERK2 accumulation requires ERK$^{T188}$-phosphorylation and impacts on hypertrophic gene regulation. a** Images and quantification of cell size of phalloidin-stained NRCM transduced with Flag-tagged ERK2-wt or Flag-ERK2$^{\Delta174-177,T188D}$ (ERK2-Δ4D) in response to PE (4 µM, 24 h) (*n* = 5 independent experiments and 25 cells per group and experiment; scale bar 20 µm). **b** Nuclear-to-cytosolic ratios of YFP-tagged ERK2-wt (ERK2-wt), YFP-ERK2$^{\Delta174-177}$ (ERK2-Δ4), and YFP-ERK2$^{\Delta174-177,T188D}$ (ERK2-Δ4D) in NRCM in response to PE (4 µM, 10 min). Shown are confocal images displaying distribution of YFP-tagged ERK2 constructs (yellow) and DAPI-stained cell nuclei (blue) (scale bar 10 µm). Quantification: Supplementary Fig. 4A. **c, d** NRCM transduced with Flag-tagged ERK2$^{T188D}$ (ERK2$^{T188D}$) or HA-tagged MEK1$^{SS218/222DD}$ (MEK-DD) treated with PE (4 µM, 24 h) or IGF (30pM, 24 h). **c** Representative phalloidin-stained NRCM (scale bar 25 µm) and cell size analysis (*n* = 3 independent experiments and 70–100 cells per group and experiment). **d** Representative western blots of pERK(TEY) and pERK(T188) were reproduced three times with similar results. **e** Nuclear-to-cytosolic ratios of NRCM transduced with YFP-tagged ERK2-wt and MEK-DD if indicated and stimulated with PE (4 µM, 10 min). Shown are confocal images of YFP-ERK2, DAPI-stained cell nuclei (scale bar 10 µm), and quantitative analyses (*n* = 30–70 cells per group of four independent experiments). **f** NRCM transduced with Flag-tagged ERK2$^{T188D}$ or the HA-tagged MEK1$^{SS218/222DD}$ mutant treated with PE (4 µM, 24 h) or IGF (30pM, 24 h). mRNA expression of connective tissue growth factor (*Ctgf*), transforming growth factorβ1 (*Tgfb1*), transforming growth factorβ2 (*Tgfb2*), transforming growth factorβ3 (*Tgfb3*), Toll-like receptor2 (*Tlr2*), ETS domain-containing protein Elk-1 (*Elk1*), insulin-like growth factor (*Igf1*), sarcoplasmic reticulum calcium ATPase 2A (*Serca2a*), peroxisome proliferator-activated receptor gamma coactivator 1-α (*Ppargc1a*), nuclear factor erythroid 2-related factor 2 (*Nrf2*), heart- and neural crest derivatives-expressed protein 2 (*Hand2*), ATP synthase, H$^+$-transporting, mitochondrial F1F0 complex, subunit E (*Atp5k*), myocyte-specific enhancer factor 2A (*Mef2a*), E3 ubiquitin-protein ligase TRIM63 (*Murf*), and cytochrome C (*CytoC*) normalized to glycerinaldehyde-3-phosphate dehydrogenase (*Gapdh*). *n* represents number of samples of biologically independent experiments measured in triplicates and are indicated in bar graph. Error bars are mean ±s.e.m.; ordinary one-way ANOVA with Tukey as post hoc test was used except for **a** where Bonferroni was used as post hoc test. Source data are provided as a Source Data file.

MEK-DD – even though ERK1/2 [pERK(TEY)] were similarly activated under all conditions (Fig. 2e and Supplementary Fig. 6H). Interestingly, gene expression patterns in response to these hypertrophic triggers clearly diverged (Fig. 2f). For example, several genes associated with a rather adaptive form of cardiac hypertrophy[24] were downregulated in response to ERK2$^{T188D}$ + PE (insulin-like growth factor, *Igf1*; Peroxisome proliferator-activated receptor gamma coactivator 1-α, *Ppargc1a*; ATP synthase, H$^+$-transporting, mitochondrial F1F0 complex, subunit E, *Atp5k*), genes associated with a maladaptive form of cardiac hypertrophy were upregulated under this condition (connective tissue growth factor, *Ctgf*; transforming growth factor β1, *Tgfb1*; transforming growth factor β2, *Tgfb2*; Toll-like receptor2, *Tlr2*), whereas no significant changes were observed with MEK-DD and

IGF. Thus, the strong effect of ERK2$^{T188D}$ + PE on gene regulation correlated well with the nuclear localization of YFP-ERK2, which was not observed in cells transduced with MEK-DD. Overall, our data suggest that ERK$^{T188}$-phosphorylation and subsequent nuclear ERK localization/accumulation are key mediators of pathological/physiological gene expression and subsequently ERK-mediated pathological cardiomyocyte hypertrophy.

**EDI attenuates pERK$^{T188}$ and nuclear ERK signaling.** To target ERK$^{T188}$-phosphorylation, we next targeted the C-terminal part of ERK2 (termed ERK-dimerization inhibitory peptide "EDI") that contains the putative ERK-ERK interface (Fig. 3a; ref. [26]). Proximity ligation and co-immunoprecipitation experiments

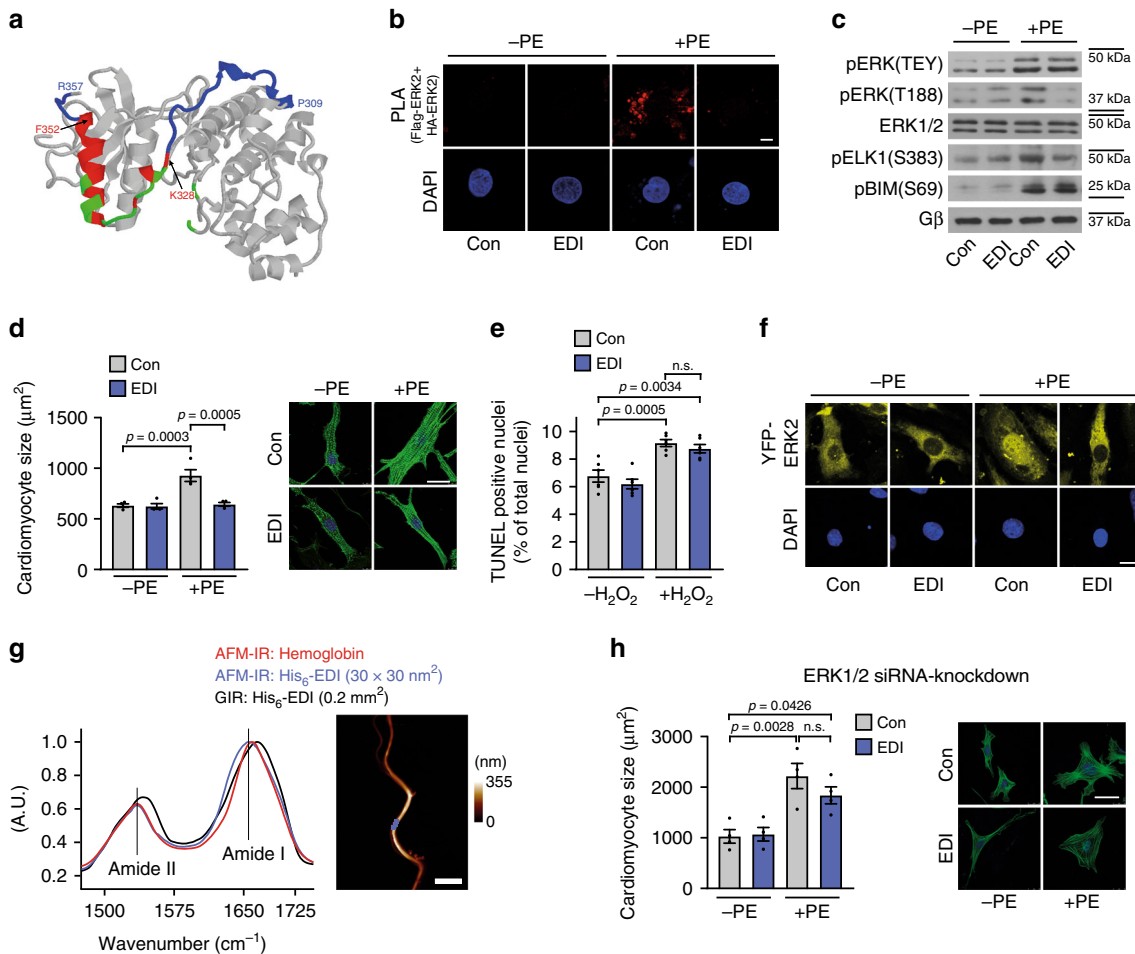

**Fig. 3 EDI-based interference with ERK dimerization. a** Proposed ERK dimer interface[26] in crystalized ERK2 (pdb ID: 2ERK). Peptide JOLU22 (ERK 328–352; red) presents nine of the 11 residues involved in dimer interactions (green). EDI (ERK 309–357) is marked in blue. **b** Proximity ligation assay (PLA) in NRCM transduced with HA- and Flag-tagged ERK2-wt and control virus (LacZ; Con) or myc-ERK2[309–357] (EDI) in response to PE (4 μM, 10 min; $n = 4$; 15–20 cells per group and experiment; scale bar 5 μm). Proximity-ligated HA- and Flag-tagged ERK are indicated in red, DAPI-stained nuclei in blue. **c–f** NRCM transduced with LacZ (Con) or EDI. **c** Representative western blots of nuclear (pELK[S383], $n = 8$) and cytosolic (pBIM[S69], $n = 13$) ERK1/2 target phosphorylation and of ERK(TEY)- and ERK(T188)-phosphorylation (pERK[TEY], $n = 8$; pERK[T188], $n = 8$) in response to PE (4 μM, 10 min). Quantifications: Supplementary Fig. 6B. **d** Representative phalloidin-stainings and quantification of cell size in response to PE (4 μM, 24 h; $n = 4$; 40 cells and experiment; scale bar 20 μm). **e** TUNEL-positive NRCM; $H_2O_2$ stimulation (100 μM, 1 h; $n = 6$ independent experiments with 500–1000 cells per experiment and group). **f** Nucleocytosolic distribution of YFP-tagged ERK2-wt (yellow) in the absence and presence of EDI and PE (4 μM, 10 min). DAPI-stained nuclei (blue). Scale bar 10 μm. Quantifications: Supplementary Fig. 4A. **g** Atomic force microscopy-infrared (AFM-IR) spectra of the purified His₆-EDI (blue) and hemoglobin (red) aggregates. The Amide I band maximum at ~1656 cm⁻¹ indicates a dominant alpha-helical structure. Hemoglobin was used as a reference protein containing α-helices. Grazing angle IR (GIR) spectrum of an ~1.5-nm-thick His₆-EDI (black), indicating a similar structure in film and aggregates. The right panel shows an AFM-image of the His₆-EDI aggregate (measured spot is marked blue, scale bar 3 μm). **h** NRCM size analyzed after treatment with siERK1/2 and Con or EDI and PE (4 μM, 24 h). Representative confocal pictures of phalloidin-stained cells and quantifications ($n = 4$; 64–80 cells per group; scale bar 20 μm). Error bars are mean ± s.e.m.; ordinary one-way ANOVA and Bonferroni as post hoc test was applied; *n* numbers represent biologically independent experiments. Source data are provided as Source Data file.

using cells overexpressing Flag- and HA-tagged ERK2-wt and HA-tagged ERK1-wt, respectively, revealed that ERK1/2 stimulation using carbachol or PE induces a stable interaction between Flag- and HA-tagged ERK hetero- or homodimers, a phenomenon that was prevented in the presence of EDI suggesting that EDI may affect both, ERK1 and ERK2, signaling (Fig. 3b and Supplementary Fig. 5A–C). EDI was N-terminally tagged with myc for visualization of successful transfection using anti-myc antibodies. Similarly, a fragment of EDI (JOLU22) presenting ERK residues 328–352, which was fused to the cell-penetrating peptide penetratin[27] and fluoresceinylated, also abolished the PE-induced interaction of HA- and Flag-tagged ERK2 in a proximity

ligation assay (Supplementary Fig. 5C). A cross-linking experiment in cells overexpressing Flag-ERK2 further revealed that EDI specifically binds to activated but not inactive, unstimulated ERK2, as suggested by the molecular weight/mobility shift of ERK2 after stimulation with carbachol (Supplementary Fig. 5D). And, as hypothesized from the experiments using monomeric ERK2-Δ4, EDI efficiently prevented ERK[T188]-phosphorylation without inhibitory impact on ERK activation (pERK[TEY]) (Supplementary Fig. 5E, F).

To evaluate whether EDI's effects on ERK-phosphorylation translate into inhibition of cardiomyocyte hypertrophy, we virally transduced NRCM with EDI (Supplementary Fig. 6A). This led to

a significant prevention of ERK$^{T188}$-phosphorylation (Fig. 3c and Supplementary Fig. 6B) and cardiomyocyte hypertrophy in response to PE (Fig. 3d). ERK activation and cell survival, however, were comparable to control conditions as observed with monomeric ERK2 (Fig. 3c, e, Supplementary Fig. 6B). Further, PE-induced cytosolic substrate phosphorylation was unaffected by EDI as shown by B-cell lymphoma 2-interacting mediator of cell death (BIM) phosphorylation[28], contrary to nuclear substrate phosphorylation as shown by significantly reduced ELK1 phosphorylation (Fig. 3c and Supplementary Fig. 6B). These selective effects of EDI on nuclear ERK effects were further validated by EDI's inhibitory effect on nuclear ERK2 accumulation (Fig. 3f and Supplementary Figs. 4A and 6C).

To test the specificity of EDI for ERK, i.e. the ERK-ERK interface[26], we performed several additional experiments: first, to ensure that EDI behaves as a structural counterpart of ERK2, we analyzed whether the isolated peptide has the potential to form a tertiary structure, i.e. an α-helix containing structure, as postulated from the crystal structure of ERK2. For these analyses, infrared spectroscopy of purified His$_6$-EDI was performed, which revealed an amide I band maximum at ~1656 cm$^{-1}$ that is characteristic for an α-helical protein[29] (Fig. 3g). Secondly, we analyzed whether the peptide is specific for the leucine zipper-like structure containing the ERK-ERK interface or if it also interferes with other leucine zipper-like structures. This was achieved by the analysis of the interaction of the transcription factors Myc and Max[30] in a proximity ligation assay. While EDI interfered with the ERK-ERK interaction, it had no impact on the PE-induced Myc-Max interaction (Supplementary Fig. 6D). Thirdly, we analyzed the impact of the EDI on cardiomyocyte hypertrophy in the absence of ERK1 and ERK2. siRNA-mediated knock-down of ERK1/2 (Supplementary Fig. 6E) abolished the anti-hypertrophic effect of EDI in PE-treated NRCM (Fig. 3h), which was not the case in the presence of endogenous ERK1/2 (Fig. 3d and Supplementary Fig. 6F). Of note, compensatory/non-ERK1/2-mediated hypertrophic signaling pathway may be responsible for cardiomyocyte hypertrophy in this experimental setting similarly as discussed for ERK1/2 knockout mouse models[31,32]. Fourthly, EDI did not prevent ERK-mediated PE-induced cardiomyocyte hypertrophy or nuclear ERK localization in the presence of an ERK$^{T188}$-phosphorylation simulating ERK2 mutant (ERK2$^{T188D}$), suggesting that EDI affects ERK1/2 upstream of ERK$^{T188}$-phosphorylation (Supplementary Fig. 6G, H).

These experiments support the specificity of EDI for the ERK-ERK interface, and reveal its function as a valuable tool to interfere with endogenous ERK$^{T188}$-phosphorylation, ERK1/2 signaling, and ultimately ERK1/2 function.

**AAV9-EDI gene therapy rescues TAC-induced heart failure**. To test EDI in a heart failure model, we subjected C57BL/6 mice to TAC surgery to induce chronic pressure overload and applied an AAV9 vector encoding EDI or enhanced green fluorescent protein (AAV9-eGFP) under the control of the CMV-enhanced myosin light chain promotor for efficient cardiac expression (Supplementary Fig. 7A)[21,33]. AAV9-EDI treatment of Wt mice resulted in a significant protection from pathological cardiac remodeling after TAC. Left ventricular wall thickness, heart weight-to-tibia length and heart weight-to-body weight ratios, and histological cardiomyocyte cross-sectional area were significantly reduced. In addition, parameters associated with cardiac function and fibrosis were improved compared to AAV9-eGFP-treated control mice, i.e. fractional shortening, left ventricular dilation, interstitial fibrosis, marker gene expression, and pulmonary congestion, and even survival of AAV9-EDI-treated

mice was slightly (not significantly) improved compared to control mice. Furthermore, apoptosis was significantly reduced in EDI-treated mice (Fig. 4a–f, Supplementary Fig. 7A and Supplementary Table 4). Of note, baseline parameters after 4 weeks of AAV9-eGFP or AAV9-EDI application were comparable with age- and gender-matched untreated control mice (Supplementary Table 5). The prominent impact of EDI on chronic pressure overload-induced changes in the heart was also reflected by gene array analysis: the top 100 regulated genes related primarily to extracellular matrix organization and structure, based on GO functional enrichment analysis (Fig. 4g and Supplementary Table 6). In addition, TAC-induced changes in gene expression categorized into functional profiles, i.e. cardiac hypertrophy, extracellular matrix, cell survival, and heart failure, were largely prevented by treatment with EDI (Supplementary Fig. 7B and Supplementary Tables 7–10).

Further, EDI-treated mice revealed a significant inhibition of ERK$^{T188}$-phosphorylation and phosphorylation of nuclear but not cytosolic ERK targets (Fig. 4h and Supplementary Fig. 7C). Interestingly, the heatmap visualization of *Nfat* and *Myc*-related genes shows that the presence of EDI prevented the activation of the related gene networks in response to TAC (Supplementary Fig. 7D and Supplementary Tables 11, 12). Both, *Nfat* and *Myc* signaling are strong triggers for cardiac remodeling processes and are enhanced by nuclear ERK[19,34,35]. These findings highlight EDI's impact on cardiac function and remodeling as well as its gatekeeper role at the nucleocytosolic interface.

**EDI reduces tumor proliferation and cardiomyocyte toxicity**. The Raf/MEK/ERK1/2 pathway is one of the most frequently dysregulated signaling pathways in cancer, in particular in melanoma, pancreatic, oral squamous cell, and colorectal cancers[36–38]. Cardiotoxicity as well as drug resistance are severe limitations of prolonged treatment with FDA approved drugs that target Raf/MEK/ERK1/2 signaling in cancer[3,39,40]. To test, whether the apparently cardio-safe EDI might also be effective in cancer and may thus represent a possible treatment strategy circumventing cardiotoxic side effects, we studied the role of ERK$^{T188}$-phosphorylation in cancer.

We found ERK$^{T188}$-phosphorylation to be strongly upregulated in colorectal and lung cancer as compared to healthy colon or lung tissue (Fig. 5a, b). EDI displayed an inhibitory effect on colon cancer cell proliferation (LS174T and HT29 cells) at least as strong as the effect of the MEK inhibitor PD98059 at concentrations previously shown to effectively reduce cancer cell proliferation[7] (Fig. 5c and Supplementary Fig. 8). Both EDI and PD98059 reduced ERK$^{T188}$-phosphorylation to a similar extent, whereas phosphorylation of the TEY motif of ERK1/2 was only inhibited by PD98059 corresponding to their effects in cardiomyocytes (Fig. 5d). These results suggest that specific interference with ERK$^{T188}$-phosphorylation may be as efficient as global kinase inhibition to attenuate cancer cell proliferation. Interestingly, AKT activation, one well-known compensatory mechanism in response to ERK1/2 inhibition in HT29 cells, was only induced by PD98059 but not by EDI, which may suggest that specific vs. global ERK1/2 inhibition may help to circumvent compensatory mechanisms to some extent (Fig. 5e)[41,42]. A gene array further revealed that the peptide strongly repressed gene expression associated with cell cycle and cell proliferation (Fig. 5f and Supplementary Tables 13–15). Expression levels of several genes were further analyzed in the colon cancer cell lines LS174T and HT29 by quantitative real-time PCR analysis: EDI was at least as efficient as PD98059 with regard to the suppression of the expression of the selected genes (Fig. 5g).

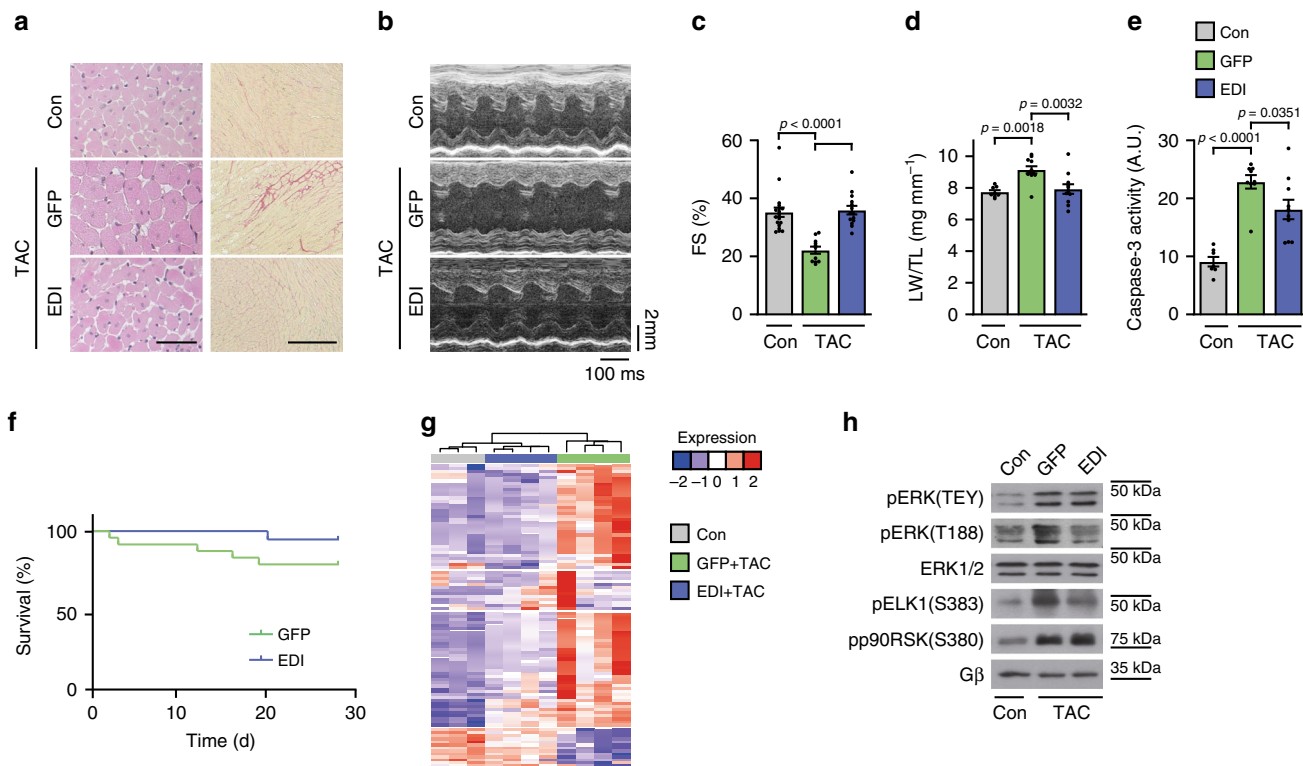

**Fig. 4 AAV9-EDI gene therapy protects from pressure overload-induced heart failure. a–h** Cardiac characterization of wild-type mice before (Con; 8-week-old C57BL/6 mice) and 4 weeks after transverse aortic constriction (TAC) and treatment with an adeno-associated virus serotype 9 (AAV9) vector encoding the peptide (EDI) or eGFP as a control (GFP) under the control of the CMV-enhanced myosin light chain promotor fragment (CMV-MLC0.26 kb). **a** Histological analyses: representative H&E (left panel; scale bar 50 μm) and Sirius Red (right panel; scale bar 200 μm)-stained pictures of left ventricular myocardial sections, quantifications are shown in Supplementary Table 4. **b** Representative M-mode echocardiograms. **c** Echocardiographic fractional shortening (Con, $n = 18$; GFP + TAC, $n = 11$; EDI + TAC, $n = 14$ mice per group). **d** Lung weight-to-tibia length (LW/TL) ratios (Con, $n = 8$; GFP + TAC, $n = 11$; EDI + TAC, $n = 11$ mice per group). **e** Caspase-3 activity in heart lysates (Con, $n = 7$; GFP + TAC, $n = 9$; EDI + TAC, $n = 10$). **f** Kaplan–Meier survival curves of AAV9-eGFP ($n = 24$) and AAV9-EDI ($n = 19$) treated mice after TAC. **g** Microarray gene expression analysis of left ventricular myocardial tissue (Con, $n = 3$; GFP + TAC and EDI + TAC, $n = 4$ mice per group). Heatmap visualization of the top 100 genes with the highest variance across the samples. For gene list and changes refer to Supplementary Table 6. **h** Representative immunoblots of heart lysates. Analysis of ERK phosphorylations (pERK[TEY], $n = 6$; pERK[T188], $n = 4$) and ERK1/2 target phosphorylation (nuclear pELK[S383], $n = 6$; cytosolic pp90RSK[S380], $n = 8$; $n$ numbers represent biologically independent experiments. For quantification refer to Supplementary Fig. 7C. Error bars are mean ± s.e.m. For statistical analyses ordinary one-way ANOVA and Bonferroni as post hoc test was applied except for **f** where a survival curve comparison was applied. Source data are provided as a Source Data file.

These findings demonstrate that $ERK^{T188}$-phosphorylation is strongly upregulated in colorectal and lung cancer and functions as a growth promoting trigger within the Raf/MEK/ERK1/2-signaling cascade in colon cancer cells. In line with these findings, our data suggest that selective inhibition of $ERK^{T188}$-phosphorylation may be advantageous in cancer therapy: this strategy efficiently attenuates cancer cell proliferation, may even circumvent compensatory mechanisms to some extent, but most importantly is cardio-safe in vitro and in mice, in contrast to PD98059 or other inhibitors of the Raf/MEK/ERK1/2 signaling cascade such as cetuximab and the clinically used MEK inhibitors trametinib, selumetinib, cobimetinib, and binimetinib. Like PD98059, all newer generation MEK inhibitors prevented $ERK^{T188}$-phosphorylation, nuclear ERK target phosphorylation and cardiomyocyte hypertrophy but also inhibited – in contrast to EDI – the phosphorylation of ERK(TEY) and of cytosolic ERK target proteins of which the Bcl2-associated agonist of cell death (Bad) is of particular importance for cell survival[43] (Fig. 6a, b and Supplementary Fig. 9A–C). The non-cardiotoxicity of EDI compared to the MEK inhibitors was validated by TUNEL assays and evaluation of mitochondrial membrane potential in response to oxidative stress; while the membrane potential was depolarized

in the presence of all MEK inhibitors, EDI protected the latter from collapsing (Fig. 6c, d vs. Fig. 3e and Supplementary Fig. 9D–F). These experiments further substantiate the essential cytosolic role of ERK1/2 signaling and the subsequent need for more specific or differential ERK1/2 targeting strategies.

## Discussion

This study reveals that interfering with ERK dimerization is highly effective in targeting pathological ERK1/2 signaling in the heart, without causing cardiotoxic side-effects. This principle may also apply more generally to unfavorable activation states characterized by increased $ERK^{T188}$-autophosphorylation, as suggested here by similar effects in cancer cells (Fig. 5). Targeting the ERK-ERK interface offers an elegant way to specifically interfere with nuclear ERK1/2 signaling. Using this peptide-based strategy, we showed (i) that ERK-ERK interaction is a prerequisite for $ERK^{T188}$-autophosphorylation of endogenous ERK1/2; (ii) that $ERK^{T188}$-phosphorylation is a central molecular event for nuclear ERK localization and signaling (Fig. 2) since simulation of $ERK^{T188}$-phosphorylation in monomeric ERK2 (ERK2-Δ4D) enabled monomeric ERK2 to accumulate in the nucleus and to

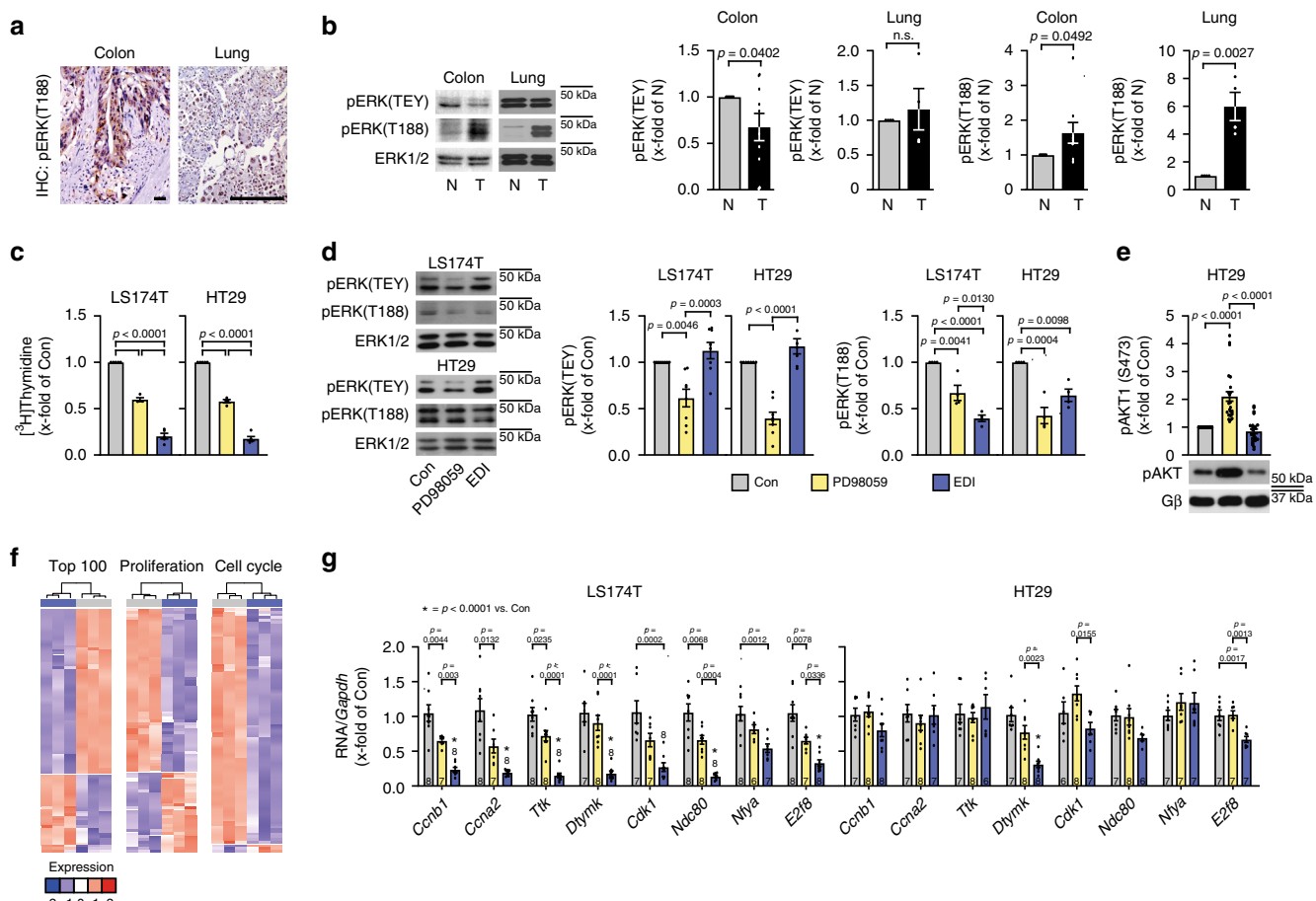

**Fig. 5 EDI reduces tumor cell proliferation. a, b** Analysis of human colon adenocarcinoma and lung cancer. **a** Immunohistochemistry (IHC) of pERK(T188) (left: colon, scale bar 50 μm, $n = 9$; right: lung, 200 μm, $n = 5$). **b** Representative western blots and quantifications of pERK[TEY] and pERK[T188]) (colon, $n = 9$ patients; lung, $n = 4$ patients; normal (N) and tumor (T) tissue). **c–g** LS174T and HT29 colon cancer cells transduced with LacZ (Con) or EDI and pretreated with PD98059 (30 μM, 48 h (**c**) or 24 h (**d**, **e**)). **c** [$^3$H]Thymidine incorporation (Con, PD98059, $n = 4$; EDI $n = 5$). **d** Respresentative western blots and quantifications of pERK(TEY) and pERK(T188) (LS174T: pERK[TEY], $n = 8$; pERK[T188], $n = 4$; HT29: pERK[TEY], $n = 7$; pERK[T188], $n = 4$). **e** Representative western blots and western blot analyses of pAKT1(S473) in HT29 cells treated as described above ($n = 28$). **f** Microarray gene expression analysis of LS174T transduced with Con or EDI ($n = 3$ per condition). Heatmaps of top 100 genes with highest variance across samples and of transcriptional changes of genes related to proliferation and cell cycle. **g** mRNA expression of cyclin B1 (*Ccnb1*), cyclin A2 (*Ccna2*), TTK Protein Kinase (*Ttk*), deoxythymidylate kinase (*Dtymk*), cyclin-dependent kinase 1 (*Cdk1*), kinetochore protein NDC80 homolog (*Ndc80*), nuclear transcription factor Y subunit alpha (*Nfya*), and E2F transcription factor 8 (*E2f8*) normalized to glycerinaldehyde-3-phosphate dehydrogenase (*Gapdh*). Error bars are mean ± s.e.m.; $n$ numbers represent biologically independent experiments; $n$ numbers of **g** are displayed in the graph; ordinary one-way ANOVA (**c–e**, **g**) and Bonferroni as post hoc test was applied except for **b** where an unpaired and two-sided Student's $t$-test was applied. Source data are provided as a Source Data file.

normally respond to hypertrophic stimuli; (iii) that ERK$^{T188}$-phosphorylation is not restricted to cardiomyocytes but is also strongly upregulated in human colon and lung cancer (Fig. 5a, b) and may thus represent a potential target and marker in cancer therapy; and (iv) that interference with nuclear ERK signaling is efficient and sufficient to interfere with maladaptive ERK1/2 signaling in the heart and in proliferating tumor cells and – most importantly – that it promises to be a cardio-safe approach to correct maladaptive, dysregulated ERK1/2 signaling (Fig. 7).

Several in part contradictory studies have shown that ERK1/2 can trigger maladaptive cardiac hypertrophy, but also physiological hypertrophy[23,44] or confer no impact on cardiac hypertrophy[1,10–13,44–50]. While it is generally accepted that ERK1/2 activation is essential for cardiomyocyte survival[45], the outcome of these mouse studies on ERK1/2-mediated cardiac hypertrophy may depend on the model, circumstances, and upstream signals. The identification of ERK$^{T188}$-autophosphorylation and molecular prerequisites for ERK$^{T188}$-phosphorylation provide additional insights into ERK1/2 signaling and in the distinct outcomes

of the mouse studies. Interestingly, only certain instances or signals upstream of ERK1/2 lead to ERK$^{T188}$-phosphorylation, and those have been shown to be rather maladaptive than physiological as physiological hypertrophy does not trigger ERK$^{T188}$-phosphorylation and its inhibition has no impact on physiological hypertrophy (ref. [10]; Supplementary Fig. 2 and Supplementary Tables 2 and 3). However, ERK$^{T188}$-phosphorylation has been identified to affect ERK localization and functional outcome. Mice with cardiac overexpression of ERK2$^{T188D}$ showed that simulation of ERK$^{T188}$-phosphorylation aggravates the pathological outcome of hypertrophic disease triggers and that overexpression of a phosphorylation-deficient mutant reduces the hypertrophic response[13]. In this study here, we made use of the peptide EDI in vitro and in vivo (AAV9-EDI gene therapy) that prevented ERK$^{T188}$-phosphorylation of endogenous ERK1/2. This "tool-peptide" modulated endogenous ERK1/2 signaling without artificial overexpression of a catalytically active protein kinase mutant and therefore provided a valuable tool to gain insights into endogenous ERK1/2 signaling. In the current study,

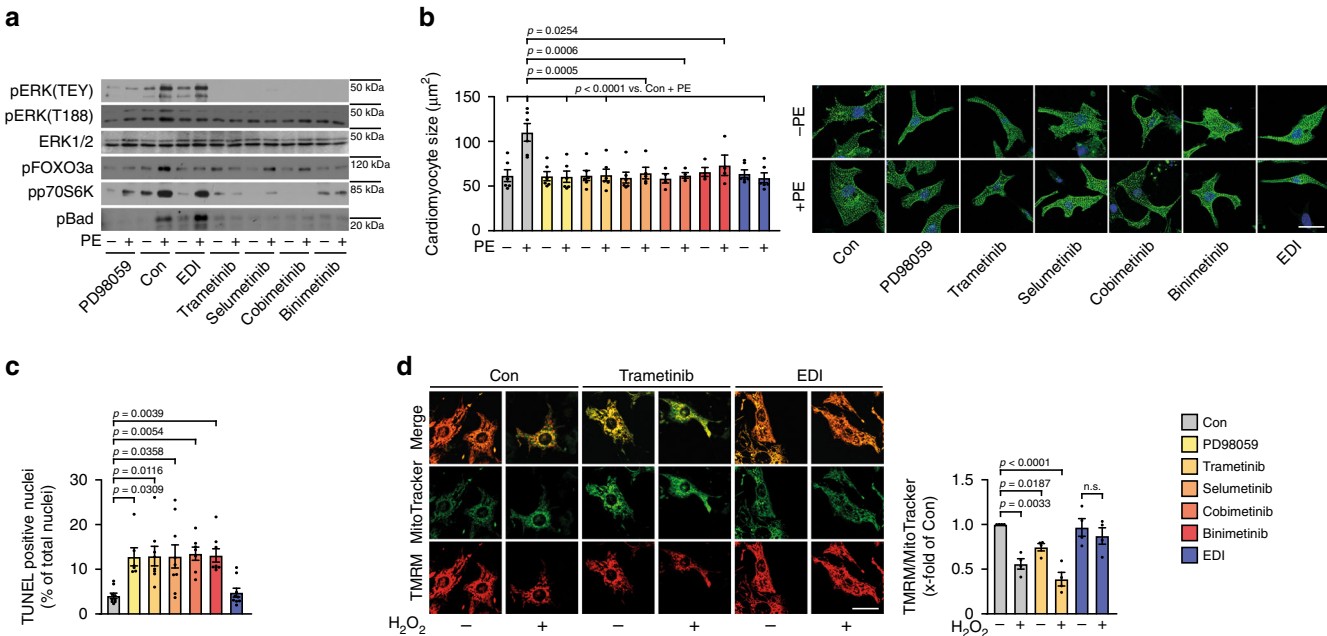

**Fig. 6 EDI protects from cardiomyocyte toxicity. a–d** NRCM transduced with LacZ (Con) or EDI and pretreated with PD98059 (30 μM), trametinib (15 μM), selumetinib (1 μM), cobimetinib (5 μM), and binimetinib (10 μM) for 1 h. **a** Representative western blots of pERK(TEY), pERK(T188), pFOXO3a (S294), pp70S6K(T421/S424), and pBad(Ser112); PE stimulation (4 μM, 10 min). Quantifications: Supplementary Fig. 9A. **b** Cardiomyocyte size in response to PE (4 μM, 24 h); phalloidin-stainings (scale bar 25 μm) and quantifications (cobimetinib-PE, cobimetinib+PE, binimetinib-PE, and binimetinib +PE $n = 4$; selumetinib+PE $n = 5$; for all other conditions $n = 6$; 50–70 cells per group and experiment). **c** TUNEL-assay (Con, selumetinib, EDI, $n = 9$; PD98059, $n = 6$; trametinib, binimetinib, $n = 8$; cobimetinib, $n = 7$, >100 cells per experiment and group). **d** Assessment of mitochondrial membrane potential in response to $H_2O_2$ (100 μM, 15 min). Tetramethylrhodamine-methyl-ester- (TMRM, red) and MitoTrackerGreen-stainings (green) (scale bar 25 μm) and quantifications (50–100 cells of $n = 4$). Error bars are mean ± s.e.m.; $n$ numbers represent biologically independent experiments; ordinary one-way ANOVA (**b**, **c**, **d**) and Tukey as post hoc test was applied. Source data are provided as a Source Data file.

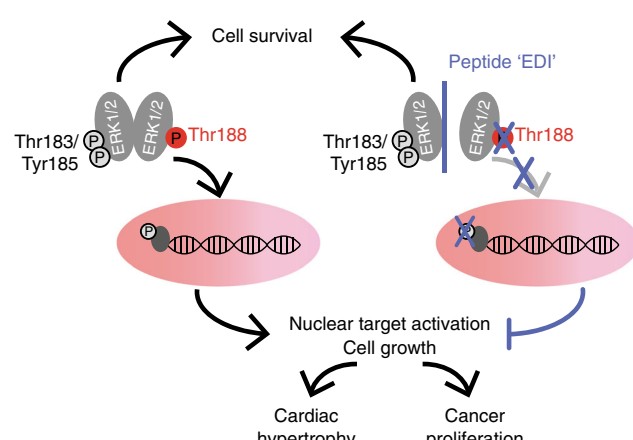

**Fig. 7 Interference with ERK-dimerization at the nucleocytosolic interface.** "EDI" interferes with ERK dimerization, thereby preventing ERK[T188]-phosphorylation and nuclear ERK accumulation and signaling without interfering with ERK1/2-mediated pro-survival signals. EDI inhibits cancer cell proliferation and maladaptive cardiac hypertrophy and does not induce cardiomyocyte apoptosis and subsequent cardiotoxicity.

we convincingly show that interference with ERK[T188]-phosphorylation indeed prevents cardiac hypertrophy and maladaptive remodeling, interferes with major hypertrophic signaling pathways[19,22,44,45] such as Myc- and NFAT-associated gene regulation in mice after TAC, and that ERK[T188]-phosphorylation is a central molecular event for nuclear ERK localization and signaling. As Crepaldi and co-workers state in their current review on ERK as a key player in the pathophysiology of cardiac hypertrophy[24], cardiac hypertrophy is a complex response to various physiological and pathological stimuli, and ERK activation seems to be involved in both adaptive and maladaptive hypertrophy, depending on the pathophysiological context. Our current and previous experiments[10,13,14] clearly suggest that pERK[T188] is a key driver of ERK1/2-mediated pathological cardiac hypertrophy. The comparison of ERK1/2 stimuli involving (ERK2[T188D] + PE) or circumventing ERK[T188]-phosphorylation (e.g. MEK-DD) revealed divergent patterns of gene regulation. In particular, ERK2[T188D] + PE modulated gene expression, i.e. suppressed genes involved in adaptive cardiac hypertrophy and induced genes involved in pathological processes (Fig. 2f). Thus, ERK[T188]-phosphorylation and subsequent nuclear ERK localization/accumulation are key determinants of gene expression and ERK-mediated pathological cardiomyocyte hypertrophy.

Selective interference with ERK[T188]-phosphorylation or ERK dimerization, in contrast to global ERK1/2 kinase inhibition and presumably also ERK kinase inhibitors[51], offers the possibility to preserve beneficial cytosolic, and to specifically interfere with maladaptive nuclear ERK1/2 signaling in cardiomyocytes. The strong effect of EDI on gene regulation and thus cardiac remodeling as well as cancer cell proliferation may be due to direct or indirect activation of nuclear transcription factors by ERK1/2, but also kinase-independent ERK1/2 effects such as a direct binding of ERK to oligonucleotides, as shown for ERK2, activated ERK2, and ERK2[T188D] (ref. 52). In line with the protection of cardiomyocyte cell death, EDI further preserved the functional integrity of cardiomyocyte mitochondria, i.e. depolarization of the mitochondrial membrane potential, in response to oxidative stress which is in clear contrast to MEK inhibition by PD98059 or new

generation MEK inhibitors (Fig. 6d and Supplementary Fig. 9F)[53]. These preserved cytosolic functions of ERK1/2 in cells transduced with EDI (vs. MEK inhibition) may be particularly relevant in patients with a second source of cardiac injury (e.g. ischemia, other toxic drugs, or hypertension). Of note, ERK1/2 knockout mice were particularly susceptible to cardiac dysfunction in response to chronic pressure overload[2,11].

A small molecule compound, DEL22379, has been described previously as a compound that binds within the ERK dimer interface similar to the peptides EDI and JOLU22 used in our study. The outcome of this compound on cancer cells, however, was different: DEL22379 efficiently induced cancer cell apoptosis, whereas our peptide displayed rather strong anti-proliferative effects as substantiated by the gene array analyses – a difference in the mode of action that needs further evaluation (Fig. 5f, g)[54,55]. Thus, slight differences in the binding mode within the ERK dimer interface result in diverse functional outcomes. These different targeting modes within the dimer interface (DEL vs. EDI) may also translate into different outcomes in neonatal cardiomyocytes with regards to cardiotoxic side-effects. In addition, long-term side-effects of EDI and the MEK inhibitors will require careful attention, as for example potential physiological functions of ERK$^{T188}$-phosphorylation are thus far unknown. Altogether, our data suggest that the ERK dimer interface is a tractable therapeutic target that may represent a "hot spot" for efficient ERK targeting. Subsequent work with in-depth analyses to better understand the potential of this hot-spot is required to further refine the herein presented anti-nuclear ERK strategy.

New targeting strategies of ERK1/2 signaling are of major therapeutic interest, since permanent activation and inactivation of this cascade both appear detrimental. Novel ERK1/2 targeting strategies should increase the therapeutic efficacy by circumventing drug-resistance and reduce cardiac side-effects that currently limit their application[2,4,56–58]. In this regard, interference with ERK dimerization holds promise, since AKT activation as one compensatory mechanism that contributes to drug-resistance was prevented in HT29 cells, and most importantly no side-effects were detectable in the AAV9-EDI model nor for the overexpression of monomeric ERK2 (Figs. 4 and 5e, Supplementary Fig. 2 and Supplementary Table 5). Thus, our targeting approach may also facilitate the application of the Raf/MEK/ERK1/2 cascade inhibitors for long-term therapy, which would be needed for the therapy of heart failure and rare genetic syndromes like RASopathies, but also with respect to chronic side effects of cancer treatment[1,13,37,47,59–63]. This targeting strategy exemplified by EDI thus may fulfill all key aspects of a potentially perfect Raf/MEK/ERK1/2 inhibitor. It will be interesting to assess whether this targeting strategy also overrides the MEK inhibitor resistance due to the recently described ERK2 mutations in human cancer, ERK2(E322K) and ERK2(D321N) located within the highly conserved common docking (CD) region close to the proposed binding site of EDI/JOLU22[64,65].

Moreover, ERK$^{T188}$-phosphorylation has important characteristics of a biomarker for certain diseases: in our study, we show for the first time that ERK$^{T188}$-phosphorylation is not restricted to pathological remodeling in the heart, but is also upregulated in cancer tissue, and thus may be central for maladaptive processes (Fig. 5)[10,45]. ERK$^{T188}$-phosphorylation was increased in 7/9 colon tumor and in 4/4 lung tumor samples, whereas pERK1/2[TEY] was rather decreased in colon cancer and unchanged in lung tumors. This is in line with the well-known variable time course of pERK1/2[TEY] due to concomitant activation of different phosphatases (Fig. 5a, b)[66,67]. Preliminary experiments suggest that ERK$^{T188}$-phosphorylation is protected from dephosphorylation within the nucleus (data not shown). ERK$^{T188}$-phosphorylation may thus become a valuable marker

for the identification of Raf/MEK/ERK1/2 participation in pathological conditions/tumors.

The identification of ERK$^{T188}$-phosphorylation as a maladaptive trigger of cardiac hypertrophy and heart failure initiated the search for a differential targeting strategy for these central kinases ERK1/2 that are involved in many physiological and pathological processes. Harnessing the molecular prerequisites for ERK$^{T188}$-phosphorylation, we here discovered a peptide/EDI-based targeting strategy of the ERK-ERK interface to interfere with ERK$^{T188}$-phosphorylation. This strategy promises to be a powerful cardio-safe treatment option to combat pathological ERK1/2-signaling in heart and cancer, and possibly other ERK1/2-related diseases requiring long-term treatments, such as heart failure and genetic syndromes.

## Methods

**Mice and rats.** Transgenic mice overexpressing ERK2$^{Δ174–177}$ (ERK2-Δ4-tg) under the control of the mouse α-myosin heavy chain (Myh6) promoter were generated by pronucleus injection of fertilized oocytes derived from FVB/N mice[13,21]. For the gene therapy approach, mice with C57BL/6J background were used. In all experiments, isogenic, age- and gender- littermates were used as controls. Pregnant Sprague Dawley rats (embryonic day 11) were purchased from Janvier. Animal care was performed corresponding to the Committee on Animal Research of the regional government (Regierung von Unterfranken and Landesamt für Natur, Umwelt und Verbraucherschutz Nordrhein-Westfalen LANUV-NRW) which reviewed and approved all experimental protocols (Az. 54–2531.01–62/06, Az. 55.2–2531.01–46/09, −20/10, −52/10, −38/11, −60/13 and −42/14 and Az. 81–02.04.2018.A082) according to the national legislation. All animals were maintained in accordance with federal guidelines: Standard chow diet and water were offered ad libitum; for housing sterilized plastic cages under specific pathogen-free conditions were used; as housing conditions 22 ± 2 °C, 12/12 light/dark cycle, 55 ± 10% humidity and <400 lux were maintained. Hygiene monitoring was done on quarterly basis.

**Human colon and lung tissue.** The use of the colon cancer samples was approved by the ethic committee of the University and University Hospital of Würzburg (20180829 01) and all patients gave written consent. Additional colon cancer and lung cancer samples were obtained from RWTH Aachen centralized Biomaterial Bank (RWTH cBMB). The cBMB was reviewed and approved by the Ethics Committee of the Medical Faculty of the RWTH Aachen University. A mandatory prerequisite for incorporation of a biomaterial sample into RWTH cBMB is the written consent of the donor. Before signing, the donor is informed by a medical doctor about the research project and the intended storage of donated samples and associated data. The important contribution of the donor to biomedical research is addressed (quoted from https://www.cbmb.rwth-aachen.de/en/data-privacy). All procedures performed involving human tissue were in accordance with the ethical standards of the institutional research committee which are comparable with the 1964 Helsinki declaration and its later amendments.

**Preparation and handling of NRCM.** Neonatal rat cardiomyocytes (NRCM) were isolated from 1–2-day-old Sprague Dawley rats[10,13,14]. NRCM were cultured in Medium Eagle (MEM) containing 5% (V/V) fetal calf serum (FCS), 100 U/mL penicillin, 100 µg/mL streptomycin, 2 mM L-glutamine, 350 mg/L NaHCO$_3$, 30 mg/L 5-bromo-2′-deoxyuridine (BrdU), and 2 mg/L vitamin B12 at 37 °C and 1% CO$_2$. FCS concentration was reduced to 1% (V/V) after 24 h followed by adenoviral transduction after 30–48 h. Cells were transduced with indicated adenoviruses encoding mouse Flag-ERK2$^{WT}$ (ERK2-wt), HA-ERK1$^{WT}$ (HA-ERK1), Flag-ERK2$^{Δ174–177}$ (ERK2-Δ4), Flag-ERK2$^{Δ174–177,T188D}$ (ERK2-Δ4D), HA-ERK2$^{T188D}$ (ERK2-D), HA-MEK1$^{SS218/222DD}$ (MEK-DD), YFP-ERK2$^{WT}$ (YFP-ERK2-wt), YFP-ERK2$^{Δ174–177}$ (YFP-ERK2-Δ4), YFP-ERK2$^{Δ174–177, T188D}$ (YFP-ERK2-Δ4D), myc-ERK2$^{309–357}$ (peptide, "EDI"), β-galactosidase (LacZ), or enhanced green fluorescent protein (GFP), respectively[10,13,14]. 48 h after transduction, NRCM were treated with either phenylephrine (PE; 4 µM, 10 min, 24 h or for the isoleucine incorporation assays for 30 h or for TUNEL assays 30 µM, 15 min) or hydrogen peroxide (H$_2$O$_2$; 100 µM, 1 h or 15 min)[10]. Before stimulation, cells were cultured in serum-starved medium. When indicated, cells were pre-incubated with PD98059 (30 µM, 1 h) trametinib (15 µM), selumetinib (1 µM), cobimetinib (5 µM), binimetinib (10 µM) for 1 h or cetuximab (0.2 µg/µl, 24 h). NRCM used in Fig. 3h and Supplementary Fig. 6E, F are cardiomyocytes purchased from Lonza (rat cardiac myocytes, R-CM-561). They were transfected with siRNAs directed against ERK1 and ERK2 or respective scramble siRNA from Dharmacon using Lipofectamine RNAiMAX (Invitrogen) according to the manufacturer's instructions (ERK1: On-Target plus Smart Pool, Rat MAPK3 (50689); ERK2: sense 5′-ACAUGGAGCUGGACGACUUUU-3′ and antisense 5′-AAGUCGUCCAGCUCCAUGUUU-3′; scramble: On-Target plus Nontargeting

siRNA#1) 24 h after plating and transduced with indicated adenoviruses 5 h later. ERK1/2 expression levels were analyzed 48 h after transfection.

**Cell culture of HEK293, COS7, HT29, LS174T, and H9c2 cells.** Human embryonic kidney 293 (HEK293) and COS7 cells were cultured in Dulbecco's Modified Eagle Medium (DMEM) at 37 °C and 7% $CO_2$ as previously described and transfected using the calcium phosphate precipitation (HEK293) or the diethylaminoethyl (DEAE)-dextran (COS7) method[13,14,68]. In all, 24 h after transfection, cells were serum starved and assays were performed 40 h after transfection. LS174T cells were maintained in Roswell Park Memorial Institute medium (RPMI) 1640 and HT29 cells in McCoy's 5A medium and H9c2 in DMEM at 37 °C and 5% $CO_2$ and transduced using adenoviruses 4 h (LS174T, HT29) and 24 h (H9c2) after seeding with indicated constructs, i.e. LacZ as control or the myc-EDI. 48 h after infection, experiments were performed. When indicated, cells were treated with MEK inhibitors 1h, 24 h or 48 h as indicated. Cell culture media were supplemented with 10% (V/V) fetal calf serum, 100 U/mL penicillin, 100 μg/mL streptomycin, and 2 mM L-glutamine.

**Antibodies.** Specified antibodies to the following proteins were used for immunoblotting (IB), immunoprecipitation (IP), immunofluorescence (IF), and immunohistochemistry (IHC): phospho-Akt1(S473) (1:2000, IB; ab81283, abcam), phospho-Bad(S112) (1:1000, IB; no. 5284, Cell Signaling), phospho-BIM(S69) (1:400, IB; no. 4581, Cell Signaling), phospho-Elk1 (S383) (1:1000, IB; no. 9181, Cell Signaling), phospho-Elk1 (S383) (1:1000, IB; ab34270, abcam), phospho-ERK1/2(TEY) (1:1000, IB; no. 9101L, Cell Signaling), ERK1/2 (1:1000, IB; no. 9102, Cell Signaling), phospho-ERK1/2(T188) (1:1000, IB; 1:500 IHC; Lorenz et al., 2009 (ref. [13]) or A010-40AP, Badrilla), Flag (1:10,000, IB or 1:200, IF; IP 5.6μg/sample;; F3165, Sigma), phospho-Foxo3a(S294) (1:1000, IB; no. 5538, Cell Signaling), Gβ (1:5000, IB; sc-378, Santa Cruz Biotechnology), HA (1:5000, IB; IP 5μg/sample; MMS-101R, Covance), HA (1:500, IF; no. 3724, Cell Signaling), Max (1:200, IF; sc-197, Santa Cruz Biotechnology), c-Myc (1:1000, IB; sc-789, Santa Cruz Biotechnology), c-Myc (1:350, IF; M4439, Sigma), phospho-p70(S6) (1:1000, IB; no. 9204, Cell Signaling), and phospho-p90RSK(S380) (1:5000, IB; ab32203, abcam).

**Immunohistochemistry.** Paraffin tissue sections were dewaxed and rehydrated, incubated in methanol containing 0.3% $H_2O_2$ for 60 min and subsequently microwaved in 10 mM citric acid (pH 6.0) for 20 min at 700 W. After washing in 0.1 M phosphate buffer with 0.25% Triton X-100 (TPBS), sections were blocked in TPBS supplemented with 5% normal goat serum for 2 h at 25 °C. Primary antibody was diluted in TPBS supplemented with 5% normal goat serum overnight at 4 °C. Primary antibodies were detected with biotinylated goat antibody to rabbit IgG followed by incubation with Vectastain Elite ABC Reagent (Vector ABC Elite kit). Tissue was then rinsed and stained with diaminobenzidine (DAB)–glucose oxidase. Counterstaining of the nuclei was performed with hematoxylin. As a control, we omitted the primary antibodies. All analyses were performed in a blinded manner.

**Immunoblot analysis and immunoprecipitations.** Left ventricular tissue, cells (NRCM, HEK293, COS7, LS174T, HT29, and H9c2) or colon and lung tissue were lysed in ice-cold lysis buffer containing 1% (V/V) Triton-X-100, 5 mM EDTA, 300 mM NaCl, 50 mM Tris (pH 7.4), 20 μg/mL soybean trypsin inhibitor, 0.4 mM benzamidine, 1 mM PMSF, 50 mM NaF, 5 mM $Na_4P_3O_7$, 1 mM $Na_3VO_4$, and 1,5 mM $NaN_3$ as previously described[10,21]. For Flag-ERK2 immunoprecipitation, Flag-specific antibody (F3165; Sigma) was preincubated with protein A-sepharose (GE Healthcare) overnight under continuous rotation at 4 °C (8 μl protein A-sepharose and 5.6 μg Flag antibody per sample). The indicated lysates were incubated with precoupled protein A-sepharose (2 h, 4 °C) and afterwards beads were washed five times with PBS (20,000×g, 1 min, 4 °C). Co-immunoprecipitated HA-ERK2 was detected by immunoblot analysis using anti-HA antibody. For HA-ERK1 immunoprecipitation, HA-specific antibody (12CA5, Roche; 8 μl protein A-sepharose and 5μg HA antibody per sample) was used as described above and co-immunopecipitated Flag-ERK2 was detected using anti-Flag antibody (F3165; Sigma).

For immunoblot analyses, proteins were separated by SDS-PAGE and transferred to PVDF or nitrocellulose membranes by wet transfer[10,21]. Membrane blocking was performed for 1 h in 5% (w/V) fat-free milk dissolved in Tris-buffered saline with 0.1% (V/V) Tween-20 (blocking milk) or in 5% (w/V) BSA dissolved in Tris-buffered saline with 0.2% (V/V) NP-40; incubation with the indicated primary antibodies was performed overnight (4 °C) and thereafter with secondary antibodies conjugated with horseradish peroxidase: 1:10,000 dilutions of anti-rabbit (no. 111–035–144) or anti–mouse (no. 115–035–003) AffiniPure antibodies from Dianova, Jackson ImmunoResearch or 1:5000 dilutions of anti–goat IgG (sc-2020, Santa Cruz). Antibodies were dissolved in BSA wash buffer containing 150 mM NaCl, 50 mM Tris (pH 7.6), 0.2% (V/V) NP-40, 0.25% (w/V) BSA or in blocking milk. After antibody incubation, membranes were washed with BSA wash buffer. For detection, Pierce ECL Plus was used. To average the immunoblot experiments and to present semiquantitative results, immunoblot signals were quantified using ImageJ or Photoshop.

**Actin staining of NRCM.** After isolation, NRCM were seeded on poly-D-lysine-coated glass coverslips. In total, 48 h after adenoviral transduction (ERK2-wt, ERK2-D, ERK2-Δ4, ERK2-Δ4D, HA-MEK-DD, myc-EDI, or LacZ), cells were stimulated with PE (4 μM, 24 h) or insulin-like growth factor (IGF, 30 pM, 24 h) and pretreated with PD98059 (30 μM), trametinib (15 μM), selumetinib (1 μM), cobimetinib (5 μM), and binimetinib (10 μM) for 1 h if indicated. NRCM were fixed with 4% (w/V) paraformaldehyde for 10 min at room temperature and incubated with AlexaFluor488-labeled phalloidin. Cell nuclei were stained with 4′,6-diamidino-2-phenylindole (DAPI). Samples were analyzed using a Leica TCS SP5 confocal microscope. DAPI was excited with a diode laser at 405 nm and measured between 430 and 453 nm and AlexaFluor488 was excited at 488 nm and emission was measured between 520 and 550 nm. Images were taken in a blinded manner. For quantification of cell size, at least 40 cells per group and experiment were analyzed by computerized pixel counting in blinded fashion.

**TUNEL-assay.** For terminal deoxynucleotidyl transferase dUTP nick-end labeling (TUNEL) of cardiac tissue, sections (2 μm) were dewaxed and rehydrated, permeabilized with proteinase K and treated with RNase. Cultured NRCM were seeded on poly-D-lysine-coated glass coverslips (250,000 cells per well in 24-well plates), stimulated with $H_2O_2$ (100 μM, 1 h) or with PE (30 μM, 15 min) and pretreated with PD98059 (30 μM), trametinib (15 μM), selumetinib (1 μM), cobimetinib (5 μM), and binimetinib (10 μM) for 1 h if indicated. Cells were fixed with 4% (w/V) paraformaldehyde for 10 min at room temperature and afterwards permeabilized with 0.1% (V/V) Triton X-100 in 0.1% (w/V) sodium citrate for 7 min on ice. TUNEL assay was then performed according to the manufacturer's protocol (no. 12156792910, Roche). Samples were incubated with the TUNEL reaction mixture for 60 min at 37 °C in a dark, humidified chamber. For positive controls, samples were preincubated with DNase I (Sigma), and for negative controls, TUNEL reaction mixture lacking terminal transferase (TdT) was used. Cell nuclei were counterstained with Hoechst 33258 (Sigma). Cell membranes in tissue sections were additionally counterstained with wheat germ agglutinin. Samples were analyzed using fluorescence microscopy (Leica; DM 4000B). All measurements and analyses were performed in a blinded manner.

**Quantification of nuclear-to-cytosolic ratios of YFP-ERK2.** Quantification of nuclear-to-cytosolic ratios of YFP-ERK2 and YFP-ERK2$^{T188D}$ (YFP-ERK2-D) was performed in COS7 cells and NRCM as previously described[10,14]. Therefore, COS7 cells were seeded on coverslips, transfected with indicated DNA constructs (pcDNA3 encoding mouse YFP-ERK2, YFP-ERK2$^{Δ174-177}$ (ERK2-Δ4), YFP-ERK2$^{Δ174-177,T188D}$ (ERK2-Δ4D), YFP-ERK$^{T188D}$ (YFP-ERK2-D), or myc-ERK2$^{309-357}$ (EDI) and human α$_{1B}$-adrenergic receptor) as described above and serum-starved 24 h after transfection. Cultured NRCM were seeded on poly-D-lysine-coated glass coverslips (300,000 cells per well in 12-well plates) and transduced with adenoviruses encoding YFP-ERK2-wt, YFP-ERK2-Δ4, YFP-ERK2-Δ4D, HA-MEK-DD, and myc-ERK2$^{309-357}$ (peptide, "EDI"). In total, 40 h after transfection, cells were stimulated with phenylephrine (PE, 10 μM for COS7 and 4 μM for NRCM; 10 min) and thereafter fixed using 4% (w/V) paraformaldehyde for 10 min at room temperature. Nuclei were stained with DAPI (700 ng/mL). For examination of nuclear-to-cytosolic ratios of YFP-ERK2 localization, a TCS SP5 confocal microscope (Leica Microsystems, Wetzlar, Germany) was used. Yellow fluorescent protein (YFP) was excited at 488 nm and DAPI at 405 nm. Fluorescence images were taken at 530–620 and 430–463 nm, respectively. Confocal images were recorded using the sequential mode option and analyzed with ImageJ software. All measurements and analyses were performed in a blinded manner.

**[³H]Isoleucine incorporation.** Rates of protein synthesis in NRCM were determined by [³H]isoleucine incorporation as previously described with or without prior stimulation using phenylephrine (PE, 4 μM, 24 h)[10,13,14].

**Voluntary running exercise.** At the age of 8 weeks, male wild-type mice (FVB/N) and ERK2$^{Δ174-177}$ transgenic mice (ERK2-Δ4-tg, FVB/N) were individually housed to a cage with free access to a running wheel and daily running distances were recorded[10].

**Transverse aortic constriction.** Male wild-type mice at the age of 8 weeks (FVB/N) and ERK2$^{Δ174-177}$ transgenic mice (ERK2-Δ4-tg, FVB/N) were subjected to transverse aortic constriction (TAC) using a 27-gauge needle to induce chronic left ventricular (LV) pressure overload[13]. Before TAC and 6 weeks after TAC, echocardiography and cardiac catheterization was performed. Hearts were isolated for biochemical, histological, and weight analyses. For gene therapy experiments, 8-week-old male wild-type mice with C57BL/6J background were used and randomly assigned to the treatment groups. AAV9 vectors were intravenously injected into the tail vein as a 150-μl bolus ($1 \times 10^{12}$ total virus particles) after TAC surgery or of control mice without surgery. Before TAC and 4 weeks after surgery, echocardiography was performed and mice were killed for indicated analyses. The surgeon was blinded to treatment groups. Mice were excluded from the study if aortic pressure gradient was below 60 mmHg. All experiments were performed in a blinded manner.

**Echocardiography**. Transthoracic echocardiograms were performed in a blinded manner using the Vevo700 or Vevo3100 high-resolution imaging systems (VisualSonics) and a 30-MHz probe. Values for end-diastolic septal and posterior wall thicknesses as well as end-diastolic and -systolic internal diameters were obtained from 2D M-mode images in the short axis view at the proximal level of the papillary muscles. Peak blood flow velocities at the site of TAC (vmax [millimeters per second]) were gained by pulsed-waved Doppler measurements. Fractional shortening (FS), ejection fraction (EF), and aortic pressure gradients (mmHg) were calculated by VisualSonics Cardiac Measurements software. Data are presented as averages of at least six cardiac cycles per animal. All measurements and analyses were performed in a blinded manner. For basal measurements a sample size of at least $n = 8$ and for treated mice $n = 9$ (RUN or TAC surgery) was chosen based on prior knowledge of statistical power from previously published experiments[9,12,13,19].

**Histological and morphometric analysis**. For histological analysis, mouse hearts were fixed in 4% (V/V) paraformaldehyde and embedded in paraffin. Tissue sections (2 μm) were either stained with hematoxylin/eosin or with Sirius Red[13]. For determination of cross-sectional areas, 50–80 individual cells per mouse and 6–10 animals per group were analyzed by computerized pixel counting. Only nucleated cardiac myocytes of transverse myocyte sections showing a centrally localized nucleus were included in the analysis. For quantification of fibrosis, Sirius Red stained sections of 6–10 animals per group and genotype were analyzed per semi-automated image analysis. All measurements were performed in a blinded manner.

**Left ventricular catheterization**. To measure left ventricular pressures, a 1.4-F pressure catheter (Millar Instruments) was inserted into the right carotid artery and advanced to the left ventricle. Dobutamine was infused in increasing doses (75, 150, 375, 750, and 1500 ng/min) into the jugular vein. Data were analyzed using the Chart software (Chart5.4, AD Instruments) as previously described[13,21]. Eight-week-old male mice (FVB/N) or mice after 6 weeks of TAC were used for the analyses. For dobutamine concentration-response curves, a sample size of at least $n = 8$ was chosen based on prior knowledge of statistical power from previously published experiments[13,21]. Catheterization and analysis were performed in a blinded fashion.

**Caspase-3 activity**. Caspase-3 activity was determined using the Caspase-Glo 3/7 Assay kit (Promega). Frozen heart samples were homogenized in lysis buffer (see above) and then centrifuged for 10 min (25,200 g). Protein concentration was adjusted to 0.2 mg/mL; 50 μl of sample and 50 μl of reaction reagent were mixed in a white 96-well plate. After 1 h of incubation, luminescence of each sample was measured using the Perkin-Elmer EnVision 2104 Multilabel Reader.

**RNA preparation and real-time PCR**. RNA from left ventricles, NRCM, H9c2, HT29, and LS174T cells was isolated using the RNeasy Kit (Quiagen) or peqGOLD TriFast™ solution (Peqlab) and reverse-transcribed using Superscript II reverse transcriptase (Invitrogen). For quantitative real-time polymerase chain reaction (RT-PCR) the C1000 Thermal Cycler CFX96 (Bio-Rad) was used and data were analyzed according to the $2^{-\Delta\Delta Ct}$ method[69]. For cDNA amplification of genes encoding brain natriuretic peptide (Nppb), collagen type III alpha 1 (Col3a1), myc-EDI (ERK2[309−357]), kinetochore protein NDC80 homolog (Ndc80), deoxythymidylate kinase (Dtymk), cyclin-dependent kinase 1 (Cdk1), TTK Protein Kinase (Ttk), cyclin B1 (Ccnb1), cyclin A2 (Ccna2), E2F transcription factor 8 (E2f8), nuclear transcription factor Y subunit alpha (Nfya), connective tissue growth factor (Ctgf), transforming growth factorβ1 (Tgfb1), transforming growth factorβ2 (Tgfb2), transforming growth factorβ3 (Tgfb3), Toll-like receptor2 (Tlr2), ETS domain-containing protein Elk-1 (Elk1), Insulin-like growth factor (Igf1), Sarcoplasmic reticulum calcium ATPase 2A (Serca2a), Peroxisome proliferator-activated receptor gamma coactivator 1-α (Ppargc1a), Nuclear factor erythroid 2-related factor 2 (Nrf2), Heart- and neural crest derivatives-expressed protein 2 (Hand2), ATP synthase, H+-transporting, mitochondrial F1F0 complex, subunit E (Atp5k), Myocyte-specific enhancer factor 2A (Mef2a), E3 ubiquitin-protein ligase TRIM63 (Murf) and Cytochrome C (CytoC) and glyceraldehyde-3-phosphate dehydrogenase (Gapdh), a reaction mixture containing SsoFast EvaGreen Supermix (Bio-Rad), and following primers were used:

*Atp5k* (rat) forward primer 5′-CTGATCCTCGGCATGGCATA-3′;
*Atp5k* (rat) reverse primer 5′-TCCTCCGCTGCTATTCTTCT-3′;
*Ctgf* (rat) forward primer 5′-CGAAGTGAGAACCGTGTGTC-3′;
*Ctgf* (rat) reverse primer 5′-CTGGCATCTCCACTCTTCCA-3′;
*CytoC* (rat) forward primer 5′-AATGGGAGATGCTGAAGCAGGCA-3′;
*CytoC* (rat) reverse primer 5′-TGGCCCTGTCTTGTGCTTCC-3′;
*Gapdh* (rat) forward primer 5′-AAGATGGTGAAGGTCGGTG -3′;
*Gapdh* (rat) reverse primer 5′-GCTTCCCATTCTCAGCCTTG-3′;
*Elk1* (rat) forward primer 5′-CACTGGAAAGCCAGGAACAC-3′;
*Elk1* (rat) reverse primer 5′-GCTGCAGGGACTGTATTGTG-3′;
*Hand2* (rat) forward primer 5′-CAGCTACATCGCCTACCTCA-3′;
*Hand2* (rat) reverse primer 5′-TTCTTGTCGTTGCTGCTCAC-3′;
*Igf1* (rat) forward primer 5′-AGGAGGGTGCAACATCAGAA-3′;
*Igf1* (rat) reverse primer 5′-GTGAACGATCCTGGGAGCTA-3′;

*Mef2a* (rat) forward primer 5′-GTGACCTAAGGCTTCCTGGT-3′;
*Mef2a* (rat) reverse primer 5′-TACACACACTCACACCGACA-3′;
*Murf* (rat) forward primer 5′-AACCTCTGCCGGAAGTGTGC-3′;
*Murf* (rat) reverse primer 5′-CACCGCGGTTGGTCCAGTAG-3′;
*Nrf2* (rat) forward primer 5′-ATTGATCCAGATGCCCACCA-3′;
*Nrf2* (rat) reverse primer 5′-TCTCGTCACTTGCTCTGAGG-3′;
*Ppargc1a* (rat) forward primer 5′-CCCAAAGGATGCGCTCTCGT-3′;
*Ppargc1a* (rat) reverse primer 5′-TTGCGATGTGTGCGGTGTCT-3′;
*Serca2a* (rat) forward primer 5′-TTCCATCTGCCTGTCCATGT-3′;
*Serca2a* (rat) reverse primer 5′-CTAGCCCGATGACTGGAAGT-3′;
*Tgfb1* (rat) forward primer 5′-TGCTTCAGCTCCACAGAGAA-3′;
*Tgfb1* (rat) reverse primer 5′-TCCAGGCTCCAAATGTAGGG-3′;
*Tgfb2* (rat) forward primer 5′-GCCCACTTTCTACAGACCCT-3′;
*Tgfb2* (rat) reverse primer 5′-AGACCCTGAACTCTGCCTTC-3′;
*Tgfb3* (rat) forward primer 5′-CTTCTGCCTTCCTTCTTGGC-3′;
*Tgfb3* (rat) reverse primer 5′-GGGAAAGATGTCTGATGCC -3′;
*Tlr2* (rat) forward primer 5′-AGACTCTGGAAGCAGGTGAC-3′;
*Tlr2* (rat) reverse primer 5′-CCTGGTGACACTCCAAGACT-3′;
*Gapdh* (mouse) forward primer 5′-TGGCAAAGTGGAGATTGTTG-3′;
*Gapdh* (mouse) reverse primer 5′-CATTATCGGCCTTGACTGTG-3′;
*Nppb* (mouse) forward primer 5′-GGATCGGATCCGTCAGTCGTT-3′;
*Nppb* (mouse) reverse primer 5′-AGACCCAGGCAGAGTCAGAAA-3′;
*Col3a1* (mouse) forward primer 5′-AAACAGCAAATTCACTTACAC-3′;
*Col3a1* (mouse) reverse primer 5′-ACCCCCAATGTCATAGG-3′;
Myc-EDI (mouse) forward primer 5′-ATGGAGCAGAAGCTCATCAGC-3′;
Myc-EDI (mouse) reverse primer 5′-TCTGTATCCTGGCTGGAATCT-3′.
*Ndc80* (human) forward primer 5′-GATGACCGCTGTCCTGTCTA-3′;
*Ndc80* (human) reverse primer 3′-TGCGCTTCATGCTTATGACC-5′;
*Dtymk* (human) forward primer 5′-TTCTGCAAATCGCTGGGAAC-3′;
*Dtymk* (human) reverse primer 3′-GGTGAAGGCCACACCAGAAA-5′;
*Cdk1* (human) forward primer 5′-TCCTCTTTCTTTCGCGCTCT-3′;
*Cdk1* (human) reverse primer 3′-GGATTCACCAATCGGGTAGC-5′;
*Ttk* (human) forward primer 5′-GAGCTCCTGGCTTCATCCATA-3′;
*Ttk* (human) reverse primer 3′-CAACAAGTTGGCCCAGAACA-5′;
*Ccnb1* (human) forward primer 5′-CCACGAACAGGCCAATAAGG-3′;
*Ccnb1* (human) reverse primer 3′-GGACCTACACCCAGCAGAAA-5′;
*Ccna2* (human) forward primer 5′-CACAAGGACACCAGGAG-3′;
*Ccna2* (human) reverse primer 3′-CGGGTTCCCGGACTTCAGTA-5′;
*E2f8* (human) forward primer 5′-AGGCACTCAGCAAGGAAGGG-3′;
*E2f8* (human) reverse primer 3′-GGGTGTCACAGGAACAGGCT-5′;
*Nfya* (human) forward primer 5′-GCACCATTCTCCAGCAAGGC-3′;
*Nfya* (human) reverse primer 3′-GTGACAGTCCCTTGCCCACT-5′;
*Gapdh* (human) forward primer 5′-ATCATCCCTGCCTCTACTGG-3′;
*Gapdh* (human) reverse primer 3′-GTCAGGTCCACCACTGACAC-5′;

**RNA isolation and reverse transcriptase (RT)-PCR**. Total RNA was isolated from indicated cells using peqGOLD TriFast™ solution (Peqlab) according to the manufacturer's protocol. Isolated RNA was transcribed into cDNA using the Superscript® First-Strand Synthesis System (Invitrogen) and cDNA was amplified by PCR. PCR products were visualized in ethidium bromide containing agarose gels and the myc-EDI (ERK2[309−357]) was detected with the following primer pair: forward: 5′-ATGGAGCAGAAGCTCATCAGC-3′, reverse: 5′-TCTGTATCCT GGCTGGAATCT-3′.

**Proximity ligation assay**. NRCM were seeded on coverslips and transduced with adenoviruses encoding Flag-tagged ERK2, HA-tagged ERK2, and myc-EDI or LacZ as indicated. COS7 cells were seeded on coverslips and transfected with Flag-tagged ERK2, HA-tagged ERK2, and myc-EDI (subcloned into pcDNA3 expression vector) or empty pcDNA3 as control (Con) as indicated. In total, 24 h after transfection or transduction, cells were serum starved and 48 h after transfection or transduction, cells were stimulated with PE (4 μM, 10 min, 37 °C). Cells were fixed with 4% (w/V) paraformaldehyde for 10 min at room temperature and permeabilized with ice-cold methanol:acetone, 1:1 (2 min). Cells were then incubated with Duolink® Blocking Solution (30 min–1 h, RT) and afterwards incubated with indicated primary antibodies diluted in Duolink® Antibody Diluent (2 h, RT) and washed with Duolink® In Situ (Sigma) Wash Buffer A. Slides were incubated with PLA probe solution (1 h, 37 °C) and again washed with buffer A. Hybridization solution containing DNA ligase was added (30 min, 37 °C), samples were washed in buffer A and subsequently incubated with the amplification solution containing polymerase (100 min, 37 °C). Slides were washed with buffer B and mounted with Duolink® In Situ Mounting Medium (containing DAPI). Images were acquired using a Leica TCS SP5 confocal microscope. DAPI was excited with a diode laser at 405 nm and measured between 430 and 453 nm. PLA signals were excited at 594 nm and recorded between 604 and 656 nm. The investigators were blinded to experimental settings during data analysis.

**Cross-linking experiments**. For cross-linking experiments, HEK293 cells were transfected with Flag-tagged ERK2 and myc-EDI (myc-ERK2[309−357]) that were subcloned into pcDNA3 expression vector or empty pcDNA3 as control (Con). In

total, 24 h after transfection, cells were serum starved and after another 24 h stimulated for 10 min with 300 μM carbachol (CCH). Then, cells were collected in ice-cold PBS supplemented with protease (20 μg/mL soybean trypsin inhibitor, 0.4 mM benzamidine, and 1 mM PMSF) and phosphatase (50 mM NaF, 5 mM $Na_4P_3O_7$, 1 mM $Na_3VO_4$, and 1.5 mM $NaN_3$) inhibitors and 0.75% (w/V) PFA. After 10 min of incubation (RT), cells were centrifuged (1700×g, 3 min, RT) and 1.25 M glycine diluted in PBS was added to the cell pellet to stop the reaction. The cell suspension was centrifuged again and the pellet was lysed in lysis buffer (50 mM Tris [pH 7.4], 150 mM NaCl, 1% (V/V) NP40, 0.5% (V/V) sodium deoxycholate, 0.1% (w/V) SDS, 1 mM EDTA) supplemented with protease and phosphatase inhibitors. Lysates were sonicated, and cell debris was removed by centrifugation (14,000×g, 30 min, 4 °C). Protein complexes were separated by SDS-PAGE and visualized by western blotting.

**Dot blot analysis**. HEK293, COS7, and NRCM were transfected as described above. Left ventricular tissue of 10–14-week-old male wild-type mice intravenously treated with AAV9 vectors (s. generation of adeno-associated virus (AAV) vectors) for 7, 14, and 28 days, and cells were lysed in PBS supplemented with 1.5 mM $NaN_3$, 50 mM NaF, 5 mM $Na_4P_2O_7$, 1 mM $Na_3VO_4$, 1 mM PMSF, 20 μg/mL soybean trypsin inhibitor, and 0.4 mM benzamidine. Lysates were dotted on PVDF membranes and the myc-EDI was visualized using indicated antibodies by immunoblot analysis.

**Generation of adeno-associated virus vectors**. The generation of myc-tagged EDI and eGFP AAV-vectors was performed analogously to the previous description[21,70]. The constructs were subcloned into the self-complementary AAV vector genome plasmid pdsCMV-MLC0.26. The plasmid AAV9 vectors for expression of myc-EDI or enhanced green fluorescent protein (GFP) were generated by cotransfection of pdsCMV-MLC0.26-myc-EDI (EDI) or pdsCMV-MLC0.26-eGFP together with pDP9rs, a derivative of pDP2rs, with the AAV9 cap gene from p5E18-VD2-9, using polyethylenimine. Subsequently, vectors were harvested after 48 h, purified by iodixanol step-gradient centrifugation and quantified using real-time PCR as reported previously.

**[3H]Thymidine incorporation**. Proliferation rates of LS174T cells were determined by [3H]thymidine incorporation. Cells were seeded in 24-well plates, adenovirally transduced and treated as described above. In total, 48 h after infection, cells were labeled with [3H]thymidine (0.5 μCi/mL) for 4 h. Proteins were then precipitated with 5% (w/V) trichloroacetic acid and incorporated [3H]thymidine was measured by scintillation counting.

**Peptide design, synthesis, and fluorescein labeling**. An ERK2 dimer structure model[26] proposes the dimer interface to be located in 11 residues, nine of which are located in the ERK segment 328–352, which is part of the C-terminal peptide EDI (Fig. 3a). We reasoned that a shorter peptide presenting this ERK2 residues 328–352 (JOLU22) should be able to bind to the ERK2 and thus competitively prevent ERK2 dimerization. JOLU22 as well as the cell-penetrating peptide penetratin (KFDMELDDLPKEKLKELIFEETARF) were synthesized by Fmoc/t-Bu-based solid-phase synthesis, as previously described[71]. For conjugation of JOLU22 to pentratin, the sequence of JOLU22 was N-terminally modified with a cysteine residue, and the penetratin sequence was N-terminally bromoacetylated. Furthermore, JOLU22 was equipped with a fluorescein moiety, which was introduced by coupling of carboxyfluorescein. The two peptides were covalently linked through chemo-selective ligation between the cysteine thiol of JOLU22 and the bromoacetyl group of penetratin, forming a thioether bond.

**PLA with cell-penetrating peptide JOLU22**. COS7 cells were seeded on coverslips and transfected with the indicated ERK2-constructs as described above. In all, 24 h after transfection, cells were serum starved and another 24 h later, cells were transfected with the cell-penetrating peptide JOLU22-penetratin conjugate in HKR solution: 5 mM HEPES, 137 mM NaCl, 2.68 mM KCl, 2.05 mM $MgCl_2$, 1.8 mM $CaCl_2$, 1 g/l glucose[72]. After 30 min of incubation (37 °C, 7% $CO_2$), cells were washed twice with HKR solution and stimulated as indicated in normal growth medium. Afterwards, cells were fixed and stained with DAPI as mentioned above and fluorescein tagged proteins were detected using a Leica TCS SP5 confocal microscope. Therefore, DAPI was excited with a diode laser at 405 nm and measured between 430 and 453 nm and fluorescein was excited with an argon laser and emission was detected between 500 and 540 nm.

**Microarray pre-processing and data analysis**. Affymetrix gene expression data were pre-processed using 'affyPLM' packages of the Bioconductor Software[73]. Genes with the strongest evidence of differential expression were obtained using a linear model fit. Data obtained from eGFP-treated mice or LacZ transduced cells were used as reference. To annotate the microarrays, custom chip definition file version 22 from Brainarray based on Entrez ID's was used. A false positive rate of $\alpha = 0.05$ with false discovery rate (FDR) correction and a fold change greater 1.5 was taken as the level of significance. To unravel patterns in the gene expression data for different pathways heatmaps the 'ComplexHeatmap' package was used.

**IR measurements of the peptide EDI**. Immobilized films of recombinant His6-tagged ERK2[309–357] (His6-EDI), purified from BL21 by His-tag affinity purification[13], were prepared by immobilization of the His6-EDI from a 3.8 mg/mL solution in binding-buffer on gold substrates (Ssens, Netherlands). After immobilization, the samples were rinsed with deionized water. IR Microscopy was performed using a Bruker Hyperion 3000 FTIR microscope with a Grazing Angle Objective at spectral resolutions of 4 cm$^{-1}$ in dry-air purged environment. A photovoltaic mercury cadmium telluride detector served for maximum linearity of the detected IR signals. AFM-IR was performed using a commercially available AFM-IR setup by Anasys Instruments (nanoIR2-FS) equipped with a tunable p-polarized MIR quantum cascade laser (QCL) by Daylight Solutions (MIRcat) with a spectral resolution of 1 cm$^{-1}$ at 20 cm$^{-1}$ per second sweep rate in ambient conditions. Grazing angle IR (GIR) spectrum is displayed reversed and baseline-corrected.

**Assessment of mitochondrial membrane potential**. After isolation, NRCM were seeded on poly-L-lysine-coated glass coverslips. In all, 48 h after adenoviral transduction (LacZ or EDI), cells were stimulated with $H_2O_2$ (15 min, 100 μM) and pretreated with trametinib (1 h, 15 μM) if indicated. H9c2 cells were seeded on poly-D-lysine-coated glass coverslips. In all, 48 h after adenoviral transduction (LacZ or EDI), cells were stimulated with $H_2O_2$ (1 h, 100 μM) and pretreated with trametinib (15 μM), selumetinib (1 μM), PD98059 (30 μM), cobimetinib (5 μM), and binimetinib (10 μM) for 1 h if indicated. NRCM were stained with 1 nM TMRM (tetramethylrhodamine methyl ester) and 200 nM MitoTrackerGreen for 1 h, H9c2 were stained with 5 nM TMRM and 200 nM MitoTrackerGreen for 1 h. Samples were analyzed using a Leica TCS SP5 confocal microscope. TMRM was excited at 561 nm and emission was measured between 580 and 700 nm. MitoTrackerGreen was excited at 488 nm and emission was measured between 500 and 530 nm. For quantification at least 85 NRCM and 15 H9c2 per condition and experiment were analyzed by computerized pixel counting. The investigators were blinded to experimental settings during data analysis.

**Statistical analysis**. For statistical analyses, we used GraphPad software (San Diego, USA), $p < 0.05$ was regarded as significant. We chose the sample sizes for all groups based on study feasibility and prior knowledge of statistical power from previously published experiments[12,13,19,37,55]. All data sets showed normal distribution.

For adequate power, we generally chose a sample size of at least $n = 6$–8 for physiological experiments and at least $n = 4$ for biochemical experiments. Our results represent the mean±standard error (mean ± SEM). P-values are indicated within the graphs. We used Student's t-test analysis for two-group comparisons (two-tailed) and one-way analysis of variance analyses (ANOVA; ordinary one-way ANOVA) if more than two groups were compared. We used Bonferroni test as post-hoc test if not stated otherwise. Parametric tests were chosen only when variances between the compared groups were not significantly different.

**Reporting summary**. Further information on research design is available in the Nature Research Reporting Summary linked to this article.

## Data availability

Gene array data are available from Array Express: accession numbers: E-MTAB-8110 and 8111. Full scans of the blots are available in Supplementary Fig 10. In addition, all source data are provided as a Source Data file. Figures and Supplementary Figures have associated raw data. Raw data for additional information required to interpret, replicate, or build upon the findings of this study are available from the corresponding author upon reasonable request. A reporting summary for this article is available as a Supplementary Information file.

## Code availability

The custom codes in R that generated the findings of this study are available by authors upon request.

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

## Acknowledgements

We thank Dr. Ulrike Zabel, Dr. Elke Butt, Julia Fender and Jessica Pfeiffer for helpful discussions and Ann-Katrin Weiss, Jonas Bodmann, Nadine Yurdagül-Hemmrich, Nicole Ziegler, Edeltraut Hoffmann-Posorske, Konstanze Schättel, Marianne Babl, and Martina Fischer for excellent technical assistance. This study was supported by the German Ministry of Research and Education (BMBF; Comprehensive Heart Failure Center Würzburg), the Ministry for Innovation, Science and Research of the Federal State of North Rhine Westphalia, the Deutsche Forschungsgemeinschaft, SFB1116 (K.L.) and the Drug Discovery Hub Dortmund (DDHD). A PhD position was awarded to T.B. by the Elite Network of Bavaria within the IDK 'receptor dynamics'.

## Author contributions

A.T., T.B., C.S., S.K., M.W.H., P.G., W.S.-H., J.L., S.H., T.S., A.R., C.K., and K.L. performed experiments and analyzed data; P.N., A.W., R.K., H.-K.M.-H., A.R., N.F., J.E., D.D., A.E.-A., J.G.H., O.J.M., K.H., F.C., and A.Z. provided feedback and revised the manuscript; K.L. wrote the manuscript and designed the project.

## Competing interests

The authors declare no competing interests.

## Additional information

Angela Tomasovic[1,2,19], Theresa Brand[1,2,19], Constanze Schanbacher[1,2,19], Sofia Kramer[1], Martin W. Hümmert[1,18], Patricio Godoy[3], Wolfgang Schmidt-Heck[4], Peter Nordbeck[5], Jonas Ludwig[6], Susanne Homann[1], Armin Wiegering[7], Timur Shaykhutdinov[8], Christoph Kratz[8], Ruth Knüchel[9], Hans-Konrad Müller-Hermelink[10], Andreas Rosenwald[10], Norbert Frey[11,12], Jutta Eichler[6], Dobromir Dobrev[13], Ali El-Armouche[14], Jan G. Hengstler[3], Oliver J. Müller[11,12], Karsten Hinrichs[8], Friederike Cuello[15,16], Alma Zernecke[17] & Kristina Lorenz[1,2,5 ✉]

[1]Institute of Pharmacology and Toxicology, University of Würzburg, 97078 Würzburg, Germany. [2]Leibniz-Institut für Analytische Wissenschaften — ISAS—e.V., 44139 Dortmund, Germany. [3]IfADo-Leibniz Research Centre for Working Environment and Human Factors at the Technical University Dortmund, 44139 Dortmund, Germany. [4]Leibniz Institute for Natural Product Research and Infection Biology -Hans Knoell Institute-, 07745 Jena, Germany. [5]Comprehensive Heart Failure Center, 97078 Würzburg, Germany. [6]Department of Chemistry and Pharmacy, Friedrich-Alexander-Universität Erlangen-Nürnberg, 91058 Erlangen, Germany. [7]Department of General, Visceral, Transplant, Vascular and Pediatric Surgery, University Hospital of Würzburg, 97080 Würzburg, Germany. [8]Leibniz-Institut für Analytische Wissenschaften — ISAS—e.V., 12489 Berlin, Germany. [9]Institute of Pathology, University Hospital Aachen, RWTH Aachen, 52074 Aachen, Germany. [10]Institute of Pathology, University of Würzburg, 97080 Würzburg, Germany. [11]Department of Internal Medicine III, University of Kiel, 24105 Kiel, Germany. [12]DZHK (German Center for Cardiovascular Research), partner site Hamburg/Kiel/Lübeck, Kiel, Germany. [13]Institute of Pharmacology, West German Heart and Vascular Center, University Duisburg-Essen, 45147 Essen, Germany. [14]Department of Pharmacology and Toxicology, TU Dresden, 01307

Dresden, Germany. [15]Institute of Experimental Pharmacology and Toxicology, Cardiovascular Research Center, University Medical Center Hamburg-Eppendorf, Martinistrasse 52, 20246 Hamburg, Germany. [16]DZHK (German Center for Cardiovascular Research), partner site Hamburg/Kiel/Lübeck, Hamburg, Germany. [17]Institute of Experimental Biomedicine, University Hospital Würzburg, University of Würzburg, 97080 Würzburg, Germany. [18]Present address: Department of Neurology, Hannover Medical School, 30625 Hannover, Germany. [19]These authors contributed equally: Angela Tomasovic, Theresa Brand, Constanze Schanbacher. ✉email: lorenz@toxi.uni-wuerzburg.de

