## [Peer Review File · Nature Communications]

Reviewers' comments:

Reviewer #1 (Remarks to the Author):

The authors put forth a study whereby they limit the dimerization and autophosphorylation of ERK2 in order to interfere with its ability to cause pathological hypertrophy. A four amino acid deletion in ERK2 was created to make the protein dimerization deficient, and this construct was tested both in vitro and in vivo, showing protection against hypertrophic growth (in vitro). The authors use TAC surgery to test the ERK2 mutant in vitro. The authors undertake a thorough analysis of the function of an inhibitory peptide (EDI) which they use to limit the dimerization and autophosphorylation. The peptide is highly specific for the interface between two ERK proteins and its delivery via AAV9 can rescue the hypertrophy caused by pressure overload. The data also show that EDI's actions on ERK may be beneficial in several types of cancer.

The authors seem to suggest that ERK signaling in the cardiomyocyte nucleus is pathological, whereas its actions in the cardiomyocyte cytoplasm are homeostatic. They also highlight that the specific inhibition of ERK autophosphorylation is beneficial, whereas other forms of inhibiting ERK phosphorylation and signaling have been detrimental. The idea that there is subtlety to the phosphorylation of ERK is intriguing. However, both ERK autophosphorylation and phosphorylation by MEK signal through the nucleus for pro-hypertrophic signaling (studies showing the phosphorylation of ERKs by MEKs also use phosphorylation of ELK1 as a readout). However the authors do not discuss what specifically about ERK T188 phosphorylation is detrimental.

One of the more interesting aspects of this paper is the suggestion that ERK signaling in the cytoplasm is necessary for the normal function of the cardiomyocyte (despite this evidence being weak in their study – having only looked at the phosphorylation of p90RSK). Still the function of ERK outside of the nucleus is understudied in cardiomyocytes, and this study could bring some attention to that.

Major points:

- Terming EDI as “cardio-safe” (line 252) may be overreaching. The authors only looked at TAC 4 weeks out, and there is no discussion of the possible long-term effects of the peptide.
- What is the half life of the peptide in vivo in the heart?

-Better discussion of the differences between ERK1 and 2 needed. Why was ERK2 subject to the 4 AA deletion and not ERK1? The authors shift from talking about ERK1/2 in the introduction, to ERK2 directly. Could there be differences in function between homo and heterodimers?

- In figure 2, the authors begin using COS7 cells instead of cardiomyocytes to show the nuclear translocation. What is the reasoning behind this? ERK translocation may be different for cardiomyocytes. (In the same vein, why were COS7 cells also used for the proximity ligation assays?)

Minor points:

- Where is the quantification of the data in Figs 1D and 1E? Quantification is also missing from Fig 4A.

-Why is the cardiomyocyte size in Figs 1A, S1C, and so on in arbitrary units and not μm^2 ?

-There is a triplet shown in sup Fig1A for pERK(TEY) and ERK1/2 – which bands are ERK1 and ERK2? The same antibodies only produce a doublet in S1D and S2A.

- In Fig S1B cell are stimulated with PE for 24h, but in S1 they are treated with PE for 10m. Why the discrepancy?

Reviewer #2 (Remarks to the Author):

The studies are interesting and novel. The work has been performed with rigor and statistical analyses are sound. The concern of this reviewer is the use of PD98059. PD98059 was the first MEK1/2/5 inhibitor developed and is not clinically relevant. Thus, the authors need to recapitulate their work, where appropriate, using the clinically relevant MEK1/2/5 inhibitors trametinib, cobimetinib and binimetinib. Understanding the impact of these drugs on cardiac function will significantly enhance the impact of the paper.

Reviewer #3 (Remarks to the Author):

Tomasovic et al., identified a new molecular strategy that targets the interface of ERK dimerization by a novel, cardio-safe peptide/EDI, as a potential treatment for ERK-related diseases. The authors show that the EDI peptide is specific to the interface of ERK-ERK interaction, prevents ERK T188 phosphorylation and subsequent nuclear ERK1/2 signaling, and thus attenuated cardiomyocyte hypertrophy without causing cardiotoxicity. Also, they show anti-proliferative effects of the EDI

peptide in colon cancer. Overall this is a well-written manuscript with a novel finding of ERK1/2 targeting strategy and possible translation into the clinical realm.

The following clarifications are needed. The comments are not in order of importance:

1. The authors showed cardiomyocyte apoptosis by TUNEL assay in Fig 1B, 3E and Sup 1E in response to hydrogen peroxide, not by phenylephrine stimulation that was used for other experiments. Did the authors observe similar effects of the ERK2- Δ 4 and EDI in response to PE?

2. Why were COS7 cell used, not cardiomyocytes, for some studies such as nuclear ERK2 accumulation in Figure 2?

3. How did the authors confirm the monomeric state of the ERK2- Δ 4 or ERK1/2 after treatment with the EDI peptide?

4. In Figure 3A, it would be better if the authors show the dimeric ERK2 structure and locate the EDI sequences that correspond to the peptide that they designed. (The author can do generate symmetry mates in Pymol).

1. Based on the structure that the authors provided, the dimeric interface seems to be smaller than the EDI sequence; the interface in the white box matches only part of the peptide sequence. Please provide a more detailed statement about the rationale for the peptide design.

2. Explain how the EDI peptide differentiates affected nuclear targets from unaffected cytosolic targets. Is it really a peptide selectivity or context selectivity (such as different stimuli; for example, mitogenic stimulation)? Provide evidence for this statement.

3. The author tested the specificity of the EDI by examining other leucine zipper like structures. Have they determined whether the sequence is conserved in that area in evolution? Are there known mutations in human within this sequence?

4. The EDI had no effect on cell survival under induced pathological conditions by either PE or hydrogen peroxide, but it showed a significant reduction in cancer. What is the molecular basis for that?

5. Physiological condition vs pathological stimulus: Can the authors speculate what senses or propagates signaling cascades? Is the enzyme state triggering phosphorylation?

Reviewer #4 (Remarks to the Author):

In this manuscript Dr. Tomasovic and colleagues further examine the role of ERK(T188) autophosphorylation and ERK2 dimerization in both cardiomyocytes and cancer cells using both in vitro and in vivo approaches. The fundamental notion underpinning the project is that the beneficial (cytoprotective) and the deleterious (pro-hypertrophic) effects of ERK1/2 activation can be decoupled by preventing ERK(T188) phosphorylation. Members of this authorship group previously have explicated the cardiac functions of pERK(T188) in multiple high-profile publications (PMIDs 19060905, 23589880). The primary contribution of the present manuscript is to introduce the translational potential of a peptide that interferes with ERK(T188) phosphorylation; novel mechanistic insight is relatively modest. Nevertheless, the experimentation is extensive and the data are convincing. The manuscript is very interesting, well-reasoned, and indeed presents new findings that could have therapeutic relevance.

MAJOR

1. In a manuscript that is attempting to establish superior cardiosafety of interference of ERK2 dimerization (over conventional MEK inhibitors) it is somewhat unfortunate that PD98059 was chosen as a comparator rather than one of the MEK inhibitors in clinical use (trametinib, cobimetinib, selumetinib). The authors certainly do not need to repeat all of their experiments using another agent, but it would be very instructive to confirm that these agents abolish ERK2(T188) phosphorylation and to see how their effects on cardiomyocytes compare to EDI (vis a vis hypertrophy and cell survival).
2. The results of the TAC/AAV-EDI experiments are quite impressive and encouraging. However, if a central point of the manuscript is the cardiosafety of EDI, it will be important to see the effect of EDI alone on the heart. I do not find these controls in the current manuscript.

MINOR

1. Though cardiotoxicity does occur as an important complication of MEK-ERK inhibition, it is a bit hyperbolic to state in the Introduction that this adverse effect has precluded broader therapeutic use. Many other factors account for the (currently) relatively narrow indications.
2. The percentage of TUNEL positive nuclei in both control and TAC animals is surprisingly (non-physiologically) high. The caspase activity assay is somewhat reassuring, but some comment on this finding would be appropriate.
3. Why doesn't the phosphomimetic ERK2d4D construct itself promote nuclear translocation (and hypertrophy) if it indeed mimics T188 phosphorylation? Is T188 phosphorylation necessary but not sufficient for nuclear translocation?

4. The effect of PD98059 on pERK(T188) is more modest than on pERK(TEY). Is there biasing in the activities of MEK/ERK inhibitors? Here also it would be useful to compare with one of the agents in clinical use.

5. The siRNA experiments suggest that ERK is not necessary for PE-mediated cardiomyocyte hypertrophy. This result seems at odds with previous publications. (PMID 10984495, 12411397). Could the authors please comment?

6. The authors use PE as a typical activator of ERK(T188), however the degree of T188 phosphorylation as shown in Figure 4C is quite modest. A summary panel indicating fold-activation across multiple experiments would be helpful. That panel could also include a quantitative indication of EDI's effect on T188 activation, which is difficult to discern in the current immunoblot.

7. Is the disproportional phosphorylation of the T188 site (vs. the TEY motif site) characteristic of tumor types other than colorectal cancer? Replication of some findings in a second tumor type in which MEK-ERK activation is oncogenic (e.g. melanoma) would be even more convincing (though are not absolutely necessary).

8. Many of us are accustomed to seeing ERK(TEY) identified by the associated phosphorylation sites (ERK T202/Y204). Could the authors please add one phrase/sentence in which the equivalence is identified?

9. The authors clearly have given careful thought to the title and needn't change it, but I am not sure that their data fully support the inclusion of "at the nucleocytoplasmic interface".

Brian Jensen MD

Associate Professor of Medicine and Pharmacology

University of North Carolina School of Medicine

Point by point response to referees' comments

We are grateful for the positive comments by all Reviewers and their interest in our work. We agree that some aspects of the work had still been somewhat preliminary and at times would have benefited from additional in-depth mechanistic information; we have, therefore, carried out considerable additional experimentation, involving both *in vitro* and *in vivo* studies to further clarify the mechanisms underlying the beneficial effects of EDI, the safety of EDI as well as the advantages of EDI over the newer generation MEK/ERK inhibitors. Most importantly, we show that the class of clinically used MEK inhibitors engage in a clearly different molecular mechanism from EDI, resulting in a superiority of EDI with regards to cardiomyocyte safety. We also noted a strong upregulation of pERK^{T188} in another tumor type, which further substantiates the potential broad therapeutic implications of our findings. We interpret our data to suggest that EDI – compared to the class of newer generation MEK/ERK inhibitors – constitutes a powerful strategy with a novel mechanistic mode of action and cardio-safe profile. We hope that with our extensive new data, which further corroborate our conclusions, and the corresponding revisions to the manuscript, you will now find our work acceptable for publication in *Nature Communications*.

In the answers to the reviewers, new Figures are marked in bold. Changes in the manuscript text are marked in red.

Reviewer #1 (reviewer comments 1-10)

- 1) The authors seem to suggest that ERK signaling in the cardiomyocyte nucleus is pathological, whereas its actions in the cardiomyocyte cytoplasm are homeostatic. They also highlight that the specific inhibition of ERK autophosphorylation is beneficial, whereas other forms of inhibiting ERK phosphorylation and signaling have been detrimental. The idea that there is subtlety to the phosphorylation of ERK is intriguing. However, both ERK autophosphorylation and phosphorylation by MEK signal through the nucleus for pro-hypertrophic signaling (studies showing the phosphorylation of ERKs by MEKs also use phosphorylation of ELK1 as a readout). However, the authors do not discuss what specifically about ERK T188 phosphorylation is detrimental.**

The Reviewer is raising an important issue. We indeed believe that selective inhibition of pERK^{T188} is beneficial for the heart whereas global ERK inhibition (inhibiting ERK phosphorylation and signaling) is detrimental.

We believe that pERK^{T188} is a key trigger for nuclear ERK localization or accumulation and has pathological effects in the heart. This hypothesis is based on ERK localization studies showing that the simulation or induction of pERK^{T188} is needed for nuclear localization of ERK. In line with these studies, inhibition of pERK^{T188} by EDI or a phosphorylation-deficient mutant prevents/reduces the phosphorylation of nuclear but not of cytosolic targets. Further, overexpression of a mutant that simulates pERK^{T188} (ERK2^{T188D}) in mouse hearts led to cardiac hypertrophy in response to chronic pressure overload (TAC-model), exaggerated fibrosis, an increase in heart failure markers and a significant decrease in cardiac function (Lorenz *et al.*, *Nat Med* 2009); in contrast, EDI or overexpression of phosphorylation-deficient mutants, ERK2^{T188A} or ERK2^{T188S}, prevented cardiac hypertrophy, fibrosis, reduced heart failure markers and largely preserved cardiac function (TAC, AngII and/or Iso treatment).

The molecular events that ultimately mediate the pathological outcome of pERK^{T188} are still largely unclear. A key prerequisite for the detrimental effects of pERK^{T188}, however, clearly seems to be its nuclear localization. As supported by experiments, EDI treatment, which interferes with ERK dimerization, ERK^{T188}-phosphorylation, and nuclear ERK accumulation (Lorenz *et al.*, *Nat Med* 2009), was associated with significant alterations in gene expression in gene array analyses (Fig. 4G) and in particular reduced expression of pathological genes, i.e. genes involved in changes in extracellular matrix, hypertrophy, cell death and heart failure (Suppl. Fig. 7A and 7B). EDI in addition prevented the activation of NFAT and Myc-related genes that have been associated with enhanced nuclear ERK signaling and are strong triggers of the cardiac remodeling processes (Suppl. Fig. 7D).

To further evaluate the different outcomes of detrimental ERK-activating stimuli involving pERK^{T188} and more physiological ERK-activating stimuli without induction of pERK^{T188}, we used ERK2^{T188D}+PE as a pathological stimulus and MEK1^{SS218/222DD} (MEK-DD) and IGF (Bueno *et al.*, *The EMBO Journal* 2000; Gallo *et al.*, *Int J Mol Sci*. 2019) as adaptive/physiological ERK1/2 stimuli. MEK-DD is a constitutively active MEK1 mutant that can activate ERK1/2 independently from extracellular signals such as GPCR/G-protein activation. The insulin-like growth factor IGF has been associated with a physiological type of cardiac hypertrophy (McMullen *et al.*, *JBC* 2004). Expression

levels or concentrations of ERK2^{T188D}+PE, MEK-DD and IGF were adjusted for comparable effects on cardiomyocyte hypertrophy (**Fig. 2C**). Under these conditions, we evaluated YFP-ERK2 localization, ERK(T188) and ERK(TEY) phosphorylation, and expression of genes known to be involved in pro-hypertrophic signaling. While comparable pERK(TEY) levels were detected in response to all three hypertrophic stimuli, only ERK2^{T188D}+PE resulted in increased pERK^{T188} levels (**Fig. 2D**). In line with the induction of pERK^{T188}, YFP-ERK2-wt showed significant nuclear localization in cells treated with ERK2^{T188D}+PE but not in the presence of MEK-DD – even though ERK1/2 [pERK(TEY)] were similarly activated under all conditions (**Fig. 2E**). Interestingly, gene expression patterns in response to these hypertrophic triggers were obviously divergent (**Fig. 2F**). For example, several genes associated with a rather adaptive form of cardiac hypertrophy were downregulated in response to ERK2^{T188D}+PE (Igf1, Ppargc1a, Atp5k), genes associated with a maladaptive form of cardiac hypertrophy were upregulated under this condition (Ctgf, Tgfb1, Tgfb2, Tlr2), whereas these genes were not significantly altered in response to MEK-DD and IGF, or were only modestly upregulated, as for example Serca2a (McMullen *et al.*, J Clin Exp Pharmacol 2008; Mudd *et al.*, Nature 2008; Lipson K. *et al.*, Fibrogenesis and Tissue Repair 2012; Heger *et al.*, B J Pharm 2015; Brown *et al.*, Nature Reviews 2016; Yu *et al.*, Mediators of Inflammation 2018). Thus, the strong effect of ERK2^{T188D}+PE on gene regulation correlated well with the nuclear localization of YFP-ERK2, which was not observed in cells transduced with MEK-DD or IGF. Strikingly, in this experimental setting, neither MEK-DD nor IGF had profound effects on gene regulation. One may speculate that the effect of ERK activation in the absence of pERK^{T188} may primarily affect protein translation since it is well known that ERK can activate p70S6K, which in turn can control protein synthesis (Wang *et al.*, JBC 2001; Wang & Proud, Circ Res. 2002).

The exact or key targets of pERK^{T188} in the nucleus driving the pathological outcome, however, are not clear. Preliminary findings suggest that pERK^{T188} is protected from dephosphorylation at threonine 188 (not shown), which may affect nuclear accumulation but does not interfere with ERK kinase activity. In addition to kinase-dependent effects, nuclear pERK^{T188} accumulation may also exert kinase-independent, as it was shown that ERK can bind to DNA or influence gene expression by preventing certain protein-protein interactions in the nucleus (Rodríguez *et al.*, JCB 2010; Hu *et al.* Cell 2009; McReynolds *et al.*, Biochemistry 2016).

As Crepaldi and co-workers state in their current review on ERK as a key player in the pathophysiology of cardiac hypertrophy (Gallo *et al.*, Int J Mol Sci 2019), cardiac hypertrophy is a complex response to various physiological and pathological stimuli, and ERK activation seems to be involved in both adaptive and maladaptive hypertrophy, depending on the pathophysiological context. Our current and previous experiments (Lorenz *et al.*, Nat Med 2009, Vidal *et al.*, Cardiovasc. Res 2012 and Ruppert *et al.* PNAS 2013) clearly suggest that pERK^{T188} is a key driver of pathological cardiac hypertrophy mediated by ERK1/2.

Overall, we interpret our combined data to suggest that pERK^{T188} and subsequent nuclear ERK localization/accumulation are key mediators of pathological/physiological gene expression and subsequently ERK-mediated pathological cardiomyocyte hypertrophy. We included the additional data (**Fig. 2C-F**, page 7/8) and a paragraph in the discussion (**page 16**).

- 2) One of the more interesting aspects of this paper is the suggestion that ERK signaling in the cytoplasm is necessary for the normal function of the cardiomyocyte (despite this evidence being weak in their study – having only looked at the phosphorylation of p90RSK). Still the function of ERK outside of the nucleus is understudied in cardiomyocytes, and this study could bring some attention to that.**

We strongly agree with the Reviewer that the functions of ERK outside the nucleus have only scantily been assessed (please, also refer to the previous comment), in particular ERK-mediated effects on protein translation and on functional mitochondrial integrity, which play a key role in programmed cell death, are only incompletely understood. To further evaluate the differential targeting strategies of ERK1/2 proposed in our manuscript (MEK inhibition vs. inhibition of ERK dimerization/pERK^{T188}), we now also assessed phosphorylation of p70S6K and of Bad as cytosolic ERK1/2 targets (**Fig. 5F**, **Suppl. Fig. 9A** and **9B**). Bad is a member of the BCL-2 family located at the outer mitochondrial membrane and mediates cell death via inhibition of the anti-apoptotic BCL-X_L. Upon phosphorylation by ERK1/2 on Ser112, pBad releases its interaction partner resulting in a suppression of apoptosis (Fang *et al.*, Oncogene 1999). The ERK1/2-Bad axis functionally couples ERK1/2-mediated cell survival to mitochondria (Baines *et al.*, Circ. Res. 2002). p70S6 kinase (p70S6K) is involved in translation and can be activated by ERK1/2 (Wang *et al.*, JBC 2001). Under conditions of chronic pressure overload it participates in the hypertrophic response of the heart (Boluy *et al.*, Cardiovasc Drugs TherL 2004). Similarly, as shown for pBIM and p90RSK in the previous version

of our manuscript, EDI did not interfere with cytosolic ERK target phosphorylation, whereas PD98059 as well as clinically used MEK inhibitors (trametinib, selumetinib, cobimetinib and binimetinib) led to a significant inhibition of pp70S6K and pBad. These experiments were performed in the cardiomyocyte cell line H9c2 and in NRCM showing similar results.

To better understand the importance of the cytosolic function of ERK1/2 or the need of an alternative ERK targeting strategy (inhibition of MEK vs. EDI), respectively, we assessed the functional integrity of mitochondria by the evaluation of the mitochondrial membrane potential under control conditions and in response to oxidative stress (H₂O₂). Incubation of cardiomyocytes with MEK inhibitors (PD98059, trametinib, selumetinib, cobimetinib and binimetinib) led to a significantly stronger depolarization of the mitochondrial membrane potential compared to controls in response to H₂O₂ (**Fig. 5I** and **Suppl. Fig. 9D**) – for incubation with cobimetinib and binimetinib even in the absence of H₂O₂ –, whereas EDI protected from a collapse of the mitochondrial membrane potential. These experiments were done in H9c2 cells and for trametinib and EDI in NRCM as well. These results are in line with increased TUNEL positive cells after treatment of cardiomyocytes with MEK inhibitors which was absent with EDI. Also, the differential phosphorylation of Bad in the presence of MEK inhibition or EDI, i.e. inhibition of nuclear ERK targets only, may contribute to the pro-apoptotic effects of MEK inhibition and the beneficial effect of EDI.

Overall, our new data (included in **Fig. 5F, 5H and 5I** and **Suppl. Fig. 9A, 9B and 9D**), discussed on **page 13/14**, significantly substantiate the essential cytosolic role of ERK1/2 signaling and the subsequent need for more specific or differential ERK1/2 targeting strategies.

Major points:

3) Terming EDI as “cardio-safe” (line 252) may be overreaching. The authors only looked at TAC 4 weeks out, and there is no discussion of the possible long-term effects of the peptide.

As also raised by Reviewer #4 (comment 22), we have now included an additional assessment of cardiac side effects of AAV-EDI in the absence of TAC, which validated the cardio-safety of the alternative ERK targeting strategy of EDI (**Suppl. Table 5**). However, we agree with the Reviewer and toned down the strong statement according to the experimental set-up (**page 12**) and included an additional discussion of possible long-term side-effects of the peptide (**page 17**). This aspect is of particular importance since pERK^{T188} has thus far only been found to be associated with pathological conditions such as heart failure, maladaptive cardiac hypertrophy, faster disease progression in aortic stenosis or cancer. Thus far, physiological functions of pERK^{T188} are unknown. Current studies are trying to assess potential essential functions of pERK^{T188} on e.g. cell differentiation or embryonal development. Nevertheless, preliminary results with mice that ubiquitously express the pERK^{T188} simulating ERK mutant (ERK2^{T188D}) and the ERK mutant deficient for the autophosphorylation site (ERK2^{T188A}) did not yet reveal any adverse effects.

4) What is the half life of the peptide in vivo in the heart?

The Referee raises an important question. In the context of the additional permission for the control AAV9 experiments (reviewer #4 comment 22), we also asked for additional animals to assess EDI expression longitudinally. The dot blots (**Suppl. Fig. 7A**) show that EDI expression is already present 7 days after application and that it reaches maximum levels two weeks after application. This is consistent with previous results using self-complementary AAV-vector genomes. These genomes have a modification resulting in packaging the usually single-stranded AAV genome as a double-stranded genome, avoiding double-strand synthesis as the rate limiting step for onset of gene expression and thus allowing significant gene expression *in vivo* already after one week (McCarty *et al.*, Gene Ther. 2001). In line with the maintenance of AAV-vector genomes as stable episomal DNA moieties, peptide levels are a result from continuous *de novo* synthesis and degradation. The citation and dot blots are now included in the manuscript (**Suppl. Fig. 7A**).

5) Better discussion of the differences between ERK1 and 2 needed. Why was ERK2 subject to the 4 AA deletion and not ERK1? The authors shift from talking about ERK1/2 in the introduction, to ERK2 directly. Could there be differences in function between homo and heterodimers?

The Reviewer is raising an excellent point. We also believe that this is an important issue and have started to address it. It is well established that ERK dimerization is an important prerequisite for pERK^{T188} (Lorenz *et al.*, Nat Med. 2009) and that it is important for the *dominant-negative* or *activating* effect of the ERK2 mutants, ERK2^{T188A} or ERK2^{T188D}, respectively (Ruppert *et al.*, PNAS 2013, Fig S6C). Further, co-immunoprecipitation experiments showed that ERK2-wt can interact as efficiently with ERK2-wt, ERK2^{T188A} and ERK1-wt which helps to explain the strong effects of ERK2^{T188A} and ERK2^{T188D} on ERK1/2-signaling via their dominant-negative/activating effect on ERK1/2 regardless of the respective isoform (Ruppert *et al.*, PNAS 2013, Fig S6A). Based on these findings and analogously to the co-immunoprecipitation experiment in **Suppl. Fig. 5B**, we have now evaluated whether EDI can also interfere with heteromeric ERK1/2 dimerization. Dimerization of ERK1-ERK2 was stimulated by carbachol and the presence of EDI indeed also inhibited ERK1-ERK2 dimerization (**Suppl. Fig. 5B**) similarly as shown for ERK2-ERK2 dimerization. This result strongly suggests that EDI affects ERK1 *and* ERK2 signaling since it cannot differentiate between homo- and heterodimers of ERK.

Monomeric ERK2 (ERK2^{A174-177}) was chosen because it has already been well characterized and its monomeric state has been validated by Khokhlatchev *et al.*, Cell 1998. In addition, Wilsbacher *et al.* (Biochemistry 2006) have used ERK2^{A174-177} to delete amino acids in their proposed ERK-interface. Finally, all pre-characterizations with regards to pERK^{T188} were performed using ERK2^{A174-177}.

The primary aim of the ERK2-Δ4 mouse model was to assess whether ERK monomers are toxic for the heart, and thus, whether it would be worthwhile to search for possibilities to interfere with endogenous ERK dimerization. With EDI, we now established a tool that interferes with homo- and heterodimers of ERK (**Suppl. Fig. 5B**) and proved that it seems to be cardio-safe in the presence and absence of chronic pressure-overload (Reviewer #4, comment 22, Fig. 4, Suppl. Fig. 7 and Suppl. Table 4 and new **Suppl. Table 5**). Our approach, however, cannot exclude that monomeric ERK1 overexpression could trigger deleterious long-term effects on the heart. The role of ERK dimerization, particularly of homo- and heterodimerization, is still incompletely understood, ERK homo- and heteromers may have distinct effects in different cell types, they seem however to contribute to neoplastic effects and maladaptive cardiac hypertrophy.

We have included the additional experiment (**Suppl. Fig 5B, page 9**), an additional comment in the introduction (**page 4**) (Khokhlatchev *et al.*, Cell 1998) and a discussion addressing long-term side-effects of EDI (**page 17**).

6) In figure 2, the authors begin using COS7 cells instead of cardiomyocytes to show the nuclear translocation. What is the reasoning behind this? ERK translocation may be different for cardiomyocytes. (In the same vein, why were COS7 cells also used for the proximity ligation assays?)

Thank you for this valid comment. We agree that validation of experiments in cardiomyocytes is essential to substantiate our main findings and conclusions. Initially we have chosen COS7 cells for exploratory experiments because handling and infection of COS7 cells is much easier than cardiomyocytes and no newborn animals were needed. All key COS7 cell experiments, i.e. PLA and nucleocytoplasmic localization, have now been validated in neonatal rat cardiomyocytes (NRCM) and the outcome was similar. These new data are now included in the main **Fig. 2B, 3B, 3F** and **Suppl. Fig. 4A**. COS7 cell experiments are now shown in the Suppl. Fig. 4B, 4C and 5C.

Minor points:

7) Where is the quantification of the data in Figs 1D and 1E? Quantification is also missing from Fig 4A.

In the figure legends, we now refer to the quantifications of these experiments in the Supplementary Tables.

8) Why is the cardiomyocyte size in Figs 1A, S1C, and so on in arbitrary units and not μm^2 ?

This is indeed correct. Arbitrary units are now converted in μm^2 . These figures have now been replaced.

9) There is a triplet shown in sup Fig1A for pERK(TEY) and ERK1/2 – which bands are ERK1 and ERK2? The same antibodies only produce a doublet in S1D and S2A.

This is correct, we now marked the different ERK bands in Suppl. Fig. 1A to avoid confusion.

In Suppl. Fig. 1A, we overexpressed Flag-ERK2-Δ4 and Flag-ERK2-wt. The Western blot shows endogenous ERK1 and ERK2 and in between either Flag-ERK2-Δ4 or Flag-ERK2-wt.

In Suppl. Fig. 1E, the cells were not transduced. Only endogenous ERK1 and ERK2 are visible.

In Suppl. Fig. 2A, we overexpressed ERK2-Δ4 in transgenic mice. The construct has no Flag-tag and appears in a fatter ERK2 band.

10) In Fig S1B cell are stimulated with PE for 24h, but in S1 they are treated with PE for 10m. Why the discrepancy?

It is correct that we evaluated cardiomyocyte hypertrophy after 24 h, while analyzing the phosphorylation status after 10 min only, because we studied rapid phosphorylation events in the absence of long-term changes under “steady state” conditions. During the revision we have repeated the suggested experiment in Suppl. Fig 1B with a 24 h stimulation. Similarly, pERK^{T188} is stable for at least 24 h in NRCM after PE stimulation (**Suppl. Fig. 1B**), and pERK(TEY) was still detectable as well. The additional experiment is now included (**Suppl. Fig. 1B**).

Reviewer #2 (reviewer comment 11):

11) The studies are interesting and novel. The work has been performed with rigor and statistical analyses are sound. The concern of this reviewer is the use of PD98059. PD98059 was the first MEK1/2/5 inhibitor developed and is not clinically relevant. Thus, the authors need to recapitulate their work, where appropriate, using the clinically relevant MEK1/2/5 inhibitors trametinib, cobimetinib and binimetinib. Understanding the impact of these drugs on cardiac function will significantly enhance the impact of the paper.

We thank the Reviewer for this excellent suggestion. We had used PD98059 for the initial proof-of-principle analyses, but the use of clinically relevant MEK inhibitors such as trametinib, cobimetinib and binimetinib is clearly indispensable. To address this suggestion, we have now explored the effects of these compounds and in addition selumetinib on cardiomyocyte apoptosis and hypertrophy and functional mitochondrial integrity. We also assessed their effects on pERK^{T188} and pERK(TEY), as well as cytosolic and nuclear ERK1/2 targets in comparison to EDI/inhibition of ERK dimerization. With these clinically relevant inhibitors, we obtained similar results as with PD98059, e.g. inhibition of phenylephrine-induced cardiomyocyte hypertrophy (**Fig. 5G**) and induction of cardiomyocyte apoptosis (**Fig. 5H**). Most importantly, the induction of apoptosis with trametinib, selumetinib, cobimetinib and binimetinib was even more pronounced in response to the “stressor” phenylephrine when compared to PD98059 (**Suppl. Fig. 8H**). To further assess the “vulnerability” of cardiomyocytes, we have in addition analyzed the collapse of the mitochondrial membrane potential in the presence of the different MEK inhibitors in response to an ischemic stimulus (H₂O₂) in a ventricular cardiomyocyte cell line (H9c2). When using inhibitor concentrations that affect cardiomyocyte hypertrophy, the decrease in mitochondrial membrane potential at baseline was even stronger for binimetinib and cobimetinib. Interestingly, EDI prevented the drop of membrane potential under both basal conditions and in response to ischemic stress, unmasking a clinically relevant benefit of our alternative ERK1/2 targeting strategy that appears to be much safer in cardiomyocytes compared to the clinically used MEK inhibitors. This experiment was also performed in NRCM using trametinib, validating the results in H9c2 cells (**Fig. 5I** and **Suppl. Fig. 9D**). The differential targeting of ERK1/2 by the MEK inhibitors or EDI was shown by immunoblot analyses assessing pERK^{T188} and pERK(TEY) and cytosolic and nuclear ERK1/2 targets (**Fig. 5H**) as well as the nucleocytoplasmic distribution of YPF-ERK2 for cobimetinib and EDI (**Fig. 3F** and **Fig R2**, comment 26). Please, also refer to comment 26 of Reviewer #4 (ERK inhibitors). These new data show that the class of clinically used MEK inhibitors engage in a clearly different molecular mechanism from EDI that constitutes a powerful strategy with a novel mechanistic mode of action and cardio-safe profile.

Reviewer #3 (reviewer comment 12-20):

Tomasovic et al., identified a new molecular strategy that targets the interface of ERK dimerization by a novel, cardio-safe peptide/EDI, as a potential treatment for ERK-related diseases. The authors show that the EDI peptide is specific to the interface of ERK-ERK interaction, prevents ERK T188 phosphorylation and subsequent nuclear ERK1/2 signaling, and thus attenuated cardiomyocyte hypertrophy without causing cardiotoxicity. Also, they show anti-proliferative effects of the EDI peptide in colon cancer. Overall

this is a well-written manuscript with a novel finding of ERK1/2 targeting strategy and possible translation into the clinical realm. The following clarifications are needed. The comments are not in order of importance:

12) The authors showed cardiomyocyte apoptosis by TUNEL assay in Fig 1B, 3E (EDI) and Sup 1E (PD) in response to hydrogen peroxide, not by phenylephrine stimulation that was used for other experiments. Did the authors observe similar effects of the ERK2-Δ4 and EDI in response to PE?

As an *in vivo* model of cardiac hypertrophy, we employed the TAC model to induce chronic pressure overload, in which increases in catecholamine levels (Schneider *et al.*, Basic Res Cardiol, 2011) and oxidative stress (Takimoto *et al.*, J Clin Invest 2005) are key hallmarks. To dissect differences between the two ERK1/2 targeting strategies on cell death (MEK inhibition vs. inhibition of ERK-dimerization by EDI) in NRCM, we selected phenylephrine as trigger of hypertrophy and H₂O₂ as trigger of apoptosis and oxidative stress, respectively.

We have now performed TUNEL experiments with phenylephrine as a catecholamine stimulus (Henaff *et al.*, Mol Pharmacol, 2000). A concentration of 30 μM phenylephrine induced a significant increase in TUNEL positive cells (Fig. R1A). To assess the effect of the different targeting strategies, we used EDI, PD98059 and all MEK inhibitors in clinical use. Surprisingly, EDI prevented the induction of apoptosis in response to PE, while MEK inhibition increased the number of TUNEL positive cells. All clinically used MEK inhibitors led to a significant increase in cell death in the absence and presence of phenylephrine, with the apoptotic effect of PD98059 being less pronounced in response to phenylephrine compared to H₂O₂ (Suppl. Fig. 1F, Suppl. Fig. 8H). Thus, it appears that MEK inhibition sensitizes cardiomyocytes to apoptosis, while inhibition of ERK dimerization or EDI do not cause or even protect cardiomyocytes from apoptosis (Suppl. Fig. 8H, Fig. 4E). Also, ERK2-Δ4 did not exaggerate cardiomyocyte apoptosis under control conditions (Fig. R1B). We have included these new data in Suppl. Fig. 8H.

A

B

Figure R1

(A-B) Analysis of cardiomyocyte apoptosis. (A) Neonatal rat cardiomyocytes (NRCM, n=10) were treated with phenylephrine (PE, 30 μM, 15min) or hydrogen peroxide (H₂O₂, 100 μM, 15min). (B) Neonatal rat cardiomyocytes (NRCM, n=6) were transduced with Flag-tagged ERK2 wild-type (ERK2-wt) or the Flag-tagged monomeric ERK2 mutant, Flag-ERK2^{Δ174-177} (ERK2-Δ4), and treated with phenylephrine (PE, 30 μM,

15min). Analysis of TUNEL positive NRCM (500-1000 cells per group and experiment; *P<0.05 vs. Con or ERK2-wt). Error bars are mean±s.e.m.. For statistical analysis ANOVA was applied.

13) Why were COS7 cell used, not cardiomyocytes, for some studies such as nuclear ERK2 accumulation in Figure 2?

Thank you for this important comment. As already stated regarding a similar comment of Reviewer #1 (comment 6), we agree with you that the validation of these experiments in cardiomyocytes is essential to substantiate our findings and conclusions. All key COS7 cell experiments with EDI were repeated in neonatal rat cardiomyocytes (NRCM) and revealed a similar outcome. We had chosen COS7 cells initially for exploratory experiments since handling and transfection of COS7 cells is much easier and since no newborn rats were needed. The new data were now included in the main Figures (Fig. 2B, 3B and 3F, Suppl. Fig. 4A) and the COS7 cell experiments were moved to the Supplementary Figures (Suppl. Fig. 4B, 4C and 5C).

14) How did the authors confirm the monomeric state of the ERK2-Δ4 or ERK1/2 after treatment with the EDI peptide?

The monomeric ERK mutant ERK2-Δ4 has been published by the group of Melanie Cobb (Khokhlatchev *et al.*, Cell 1998). We have chosen this mutant since it is best characterized and has been used in our previous experiments and publications. The monomeric state was originally evaluated by classical and robust *in vitro* experiments, *i.e.* ultracentrifugation and gel filtration (Khokhlatchev *et al.*, Cell 1998). Subsequently, a co-immunoprecipitation assay

clearly showed that HA-tagged ERK2-wt can co-immunoprecipitate with Flag-tagged ERK2-wt (and T188 mutants), but not with ERK2-Δ4 (Lorenz *et al.*, Nat Med. 2009, Fig. 4i).

With regards to the peptide, we assessed the efficiency of ERK2-wt dimerization and ERK1-ERK2 dimerization in the absence and presence of the peptide by performing co-immunoprecipitation assays, which showed an efficient inhibition of HA- and Flag-tagged ERK interaction after stimulation of the cells in the presence of the peptide (**Suppl. Fig. 5B**; please also refer to Reviewer #1 comment 5). Further, we assessed the association of HA- and Flag-tagged ERK2-wt by a proximity ligation assay (PLA), which shows a prominent increase of the signal in response to phenylephrine in COS7 cells (Suppl. Fig. 5C), i.e. HA-ERK2-Flag-ERK2 in close proximity (below 40 nm), in the absence of EDI expression. In the presence of EDI the PLA signal was strongly reduced. We have now repeated this experiment in NRCM obtaining similar results as seen in COS7 cells (**Fig. 3B**).

15) In Figure 3A, it would be better if the authors show the dimeric ERK2 structure and locate the EDI sequences that correspond to the peptide that they designed. (The author can do generate symmetry mates in Pymol).

As requested by the Reviewer we have drafted a new Figure (**Fig. 3A**) showing the location of EDI, the shorter peptide JOLU22, as well as the ERK residues proposed to be involved in the dimer interface. Please, also refer to our response to the next comment.

16) Based on the structure that the authors provided, the dimeric interface seems to be smaller than the EDI sequence; the interface in the white box matches only part of the peptide sequence. Please provide a more detailed statement about the rationale for the peptide design.

We agree with the Reviewer that the dimer interface appears to be smaller than the ERK sequence presented by EDI. Therefore, we have synthesized and tested a shorter peptide on its ability to interfere with ERK dimerization using a proximity ligation assay (JOLU22, ERK residues 328-352). JOLU22 represents the part of EDI that contains the majority of residues assumed to be involved in the dimer interface. It should be noted, however, that no experimentally determined ERK dimer structures are available, and that these considerations are based exclusively on computational modeling predicted dimer structures (Ref.26). As requested by the Reviewer, we have included a more detailed statement on the rationale for the peptide design (**Fig. 3A** legend and **page 34**).

17) Explain how the EDI peptide differentiates affected nuclear targets from unaffected cytosolic targets. Is it really a peptide selectivity or context selectivity (such as different stimuli; for example, mitogenic stimulation)? Provide evidence for this statement.

Thank you for these stimulating questions. The rationale of our current concept is that pERK^{T188} is important for nuclear ERK1/2 accumulation and nuclear ERK target activation, and that EDI prevents the dimerization of ERK2 which is a prerequisite for the induction of pERK^{T188}. Thus, EDI affects ERK1/2 signaling indirectly by preventing pERK¹⁸⁸ in the presence of pERK^{T188}-stimulating triggers. To further validate this hypothesis according to the question raised by the Reviewer, we asked whether EDI would prevent nuclear localization and ERK-mediated cardiomyocyte hypertrophy if pERK^{T188} is already “present”, i.e. if simulated by an pERK^{T188} simulating ERK2 mutant (ERK2^{T188D}). These experiments revealed that EDI only prevented PE-induced hypertrophy in cardiomyocytes transduced with ERK-wt but not ERK2^{T188D}. In line with these results, EDI was also incapable of preventing PE-induced YFP-ERK2^{T188D} but not YFP-ERK-wt nuclear localization. These experiments confirm that EDI only indirectly affects ERK1/2 mediated signaling (i.e. cardiomyocyte hypertrophy and cellular ERK localization) by interference with ERK dimerization/pERK^{T188} but not by ERK-substrate interaction even though subtle structural changes caused by the binding of EDI or JOLU22 cannot be completely excluded. This could be of relevance, given that the neighboring area, the common docking (CD) site, which is important for ERK2 binding to regulators and substrates, has recently been suggested to be an energetic hotspot in ERK2 (please also refer to the next comment; Taylor *et al.*, PNAS 2019). Thus far, however, no obvious differences with regards to cytosolic ERK target activation have been observed in the absence or presence of EDI suggesting that EDI is rather specific for inhibition of pERK^{T188}/ERK-dimerization. New experiments are now included in **Suppl. Fig. 6G and 6H**. Please also refer to comment 1 by reviewer #1.

18) The author tested the specificity of the EDI by examining other leucine zipper like structures. Have they determined whether the sequence is conserved in that area in evolution? Are there known mutations in human within this sequence?

We thank the Reviewer for this intriguing question. Indeed, mutations close to this leucine zipper like structure have been described in cancer (Taylor *et al.*, PNAS 2019; Chakraborty *et al.*, Oncotarget 2017; Bott *et al.*, FEBS Letters 1994). These mutations, i.e. ERK2(E322K) or ERK2(D321N), are located within the highly conserved common docking (CD) region of ERK2 that is important for the docking of substrate and regulators. These mutations cause increased ERK activity in cells and evade inactivation by dual-specificity phosphatases, and also seem to be resistant to MEK/pathway inhibitors. It will be interesting to assess whether the alternative ERK targeting strategy by EDI will override the resistance of these ERK mutants to pathway inhibitors or if the mutations also influence EDI/JOLU22 binding to ERK. This aspect has now been included in the discussion (**page 18**).

19) The EDI had no effect on cell survival under induced pathological conditions by either PE or hydrogen peroxide, but it showed a significant reduction in cancer. What is the molecular basis for that?

This is a challenging question. In cardiomyocytes, EDI interferes with hypertrophic ERK signaling and preserves cell survival. The latter is not desirable in the context of cancer treatment. However, EDI was found to strongly interfere with the expression of cell cycle genes (Fig. 5E), which might override the potential anti-apoptotic effects of ERK1/2 in cancer cells to significantly reduce cancer cell proliferation. In addition, there are several reports of compounds targeting the Raf/MEK/ERK signaling pathway (U0126, Zhou *et al.*, Oncology Letters 2019; RGCC416, Hatzidaki *et al.*, Anti-Cancer Drugs 2019) that preferentially affect cancer cell proliferation. Indeed, even mechanisms of ERK1/2-mediated cell death have been described in cancer cells, for example involving ERK1/2 binding to DAPK (death associated protein kinase) within its death domain, which prevents nuclear ERK translocation and promotes apoptotic cell death. This mechanism would possibly suggest that mechanisms blocking nuclear ERK translocation may enhance the cell death-inducing activity of ERK1/2 and provide a better means to kill tumor cells (Mebratu & Tesfaigzi, Cell Cycle 2009; Elbadawy *et al.*, Int J Mol Sci. 2018). Thus, cellular mechanisms engaged by ERK1/2 in cancer cells and cardiomyocytes seem to involve different major players. Future studies will address this important issue of differential ERK1/2 signaling in different cell types. Additional references and a comment are included in the discussion (**page 17**).

20) Physiological condition vs pathological stimulus: Can the authors speculate what senses or propagates signaling cascades? Is the enzyme state triggering phosphorylation?

We thank the Reviewer for the important question that has also been raised by Reviewer #1. Please refer to comments 1) and 2).

Reviewer #4 (reviewer comment 21-31):

In this manuscript Dr. Tomasovic and colleagues further examine the role of ERK(T188) autophosphorylation and ERK2 dimerization in both cardiomyocytes and cancer cells using both in vitro and in vivo approaches. The fundamental notion underpinning the project is that the beneficial (cytoprotective) and the deleterious (pro-hypertrophic) effects of ERK1/2 activation can be decoupled by preventing ERK(T188) phosphorylation. Members of this authorship group previously have explicated the cardiac functions of pERK(T188) in multiple high-profile publications (PMIDs 19060905, 23589880). The primary contribution of the present manuscript is to introduce the translational potential of a peptide that interferes with ERK(T188) phosphorylation; novel mechanistic insight is relatively modest. Nevertheless, the experimentation is extensive and the data are convincing. The manuscript is very interesting, well-reasoned, and indeed presents new findings that could have therapeutic relevance.

MAJOR

21) In a manuscript that is attempting to establish superior cardiosafety of interference of ERK2 dimerization (over conventional MEK inhibitors) it is somewhat unfortunate that PD98059 was chosen as a comparator rather than one of the MEK inhibitors in clinical use (trametinib, cobmetinib, selumetinib). The authors certainly do not need to repeat all of their experiments using another agent,

but it would be very instructive to confirm that these agents abolish ERK2(T188) phosphorylation and to see how their effects on cardiomyocytes compare to EDI (vis a vis hypertrophy and cell survival).

This clinically important issue has also been raised by Reviewer #2 (comment 11). We performed extensive additional experiments with the suggested MEK inhibitors in clinical use and compared their effects to EDI and PD98059 on ERK inhibition, cardiomyocyte hypertrophy and cell survival. The results are now included in **Fig. 5F-H, Suppl. Fig. 8H and Suppl. Fig. 9** in the manuscript. Please, also refer to question 11 of Reviewer #2. Compared to PD98059, trametinib, cobimetinib, selumetinib and binimetinib had a similar efficacy of inhibition of pERK^{T188} and pERK(TEY), cardiomyocyte hypertrophy and cell survival. All additional results on clinically relevant MEK inhibitors are included in **Fig. 5F-I, Suppl. Fig. 8H and Suppl. Fig. 9**.

22) The results of the TAC/AAV-EDI experiments are quite impressive and encouraging. However, if a central point of the manuscript is the cardiosafety of EDI, it will be important to see the effect of EDI alone on the heart. I do not find these controls in the current manuscript.

Following this valid remark, we have performed experiments with animals treated with EDI and GFP alone. These treatments did not cause any maladaptive changes with regards to ejection fraction, wall thickness, cardiomyocyte hypertrophy, interstitial fibrosis and heart or lung weight compared to untreated age- and gender-matched wild-type mice. These experiments further substantiate the good tolerability of EDI. These data are now included in **Suppl. Table 5** (page 12/3).

MINOR

23) Though cardiotoxicity does occur as an important complication of MEK-ERK inhibition, it is a bit hyperbolic to state in the Introduction that this adverse effect has precluded broader therapeutic use. Many other factors account for the (currently) relatively narrow indications.

We agree with the Reviewer's assessment and reworded this paragraph accordingly (**page 4**).

24) The percentage of TUNEL positive nuclei in both control and TAC animals is surprisingly (non-physiologically) high. The caspase activity assay is somewhat reassuring, but some comment on this finding would be appropriate.

For comparison to other studies and prior own data, we converted apoptotic cell numbers from TUNEL positive cells/section to TUNEL positive cells/10⁵ cells (Fig. 1F). The number of TUNEL positive cells in wild-type mice (\pm TAC) appears to be comparable to those of other groups (Liu *et al.*, *Circ Res* 2009) and our previous data (Ruppert *et al.*, *PNAS* 2013). As the Reviewer states, it is important to use independent methods for data validation. The application of a different method of evaluating caspase activity confirmed an about doubled level of apoptotic cells/caspase activity in response to TAC and no exaggeration of apoptosis in ERK2- Δ 4 or even reduced level of apoptosis in AAV-EDI-treated animals compared to respective controls.

25) Why doesn't the phosphomimetic ERK2 Δ 4D construct itself promote nuclear translocation (and hypertrophy) if it indeed mimics T188 phosphorylation? Is T188 phosphorylation necessary but not sufficient for nuclear translocation?

This is a very interesting and important question. We believe that pERK^{T188} is needed but not sufficient for nuclear ERK accumulation. As the Reviewer suggests, simulation of pERK^{T188} by ERK2^{T188D} or ERK2 ^{Δ 4D} should otherwise be sufficient for nuclear localization. However, all our experiments with the ERK2^{T188D} mutant, show pronounced nuclear localization of ERK2^{T188D} only in response to an additional ERK-activating signal, e.g. carbachol in COS cells (Lorenz *et al.*, *Nat Med* 2009), PE in COS cells (**Suppl. Fig. 6H**) or TAC (Lorenz *et al.*, *Nat Med* 2009). Similarly, cardiomyocyte/cardiac hypertrophy was not inducible by ERK2^{T188D} alone (Lorenz *et al.*, *Nat Med* 2009). Of note, the presence of ERK2^{T188D} was sufficient to induce cardiac hypertrophy in the presence of a non-hypertrophic (but ERK1/2 activating) stimulus in mice (Lorenz *et al.*, *Nat Med* 2009). Overall, we interpret our combined data to suggest that additional signaling events and stressors are key and needed to induce nuclear ERK2^{T188D} translocation. The data in COS7 cells have been confirmed in cardiomyocytes (**Fig. 2B and Suppl. Fig. 4A**) and an additional comment for better clarification is included in the main text (**page 8**).

26) The effect of PD98059 on pERK(T188) is more modest than on pERK(TEY). Is there biasing in the activities of MEK/ERK inhibitors? Here also it would be useful to compare with one of the agents in clinical use.

[Redacted]

27) The siRNA experiments suggest that ERK is not necessary for PE-mediated cardiomyocyte hypertrophy. This result seems at odds with previous publications. (PMID 10984495, 12411397). Could the authors please comment?

This is indeed a valid question, which is difficult to answer and highly discussed in the literature. Especially, Jeff Molkenin has addressed this question using ERK1/2 knockout mice, e.g. deletion of all ERK1/2 protein in the heart (Erk1^{-/-} ERK2fl/fl-Cre), which did not block the cardiac hypertrophic response *per se*, i.e. the hearts still increased in weight in response to physiological and pathological stress stimulation (Kehat *et al.*, Circ Res. 2011). A similar phenomenon was observed in DUSP6 overexpressing mice (cardiac specific overexpression), in which all cardiac ERK1/2 activation was prevented (Maillet *et al.*, JBC 2008). These publications were/are questioning the relevance of ERK1/2 for cardiac hypertrophy. The activation of alternative or compensatory signaling pathways is most likely involved in the cardiac hypertrophy response in these mice, and, further, a unique form of hypertrophy is developed in ERK1/2-KO mice: Loss of ERK1/2 induces preferential eccentric cardiomyocyte growth (Kehat *et al.*, Circ Res. 2011). Thus, alternative signaling pathways seem to be able to take over the hypertrophic response. Thus, ERK1/2 does not seem to be essential for the development of cardiac hypertrophy. In contrast, however, ERK1/2 are involved in the induction of pathological cardiac hypertrophy under certain conditions, and α -MHC-ERK2^{T188D} transgenic mice impressively revealed the detrimental effects of ERK1/2 signaling (involving pERK^{T188}) in the heart. A similar condition was used in our experimental *in vitro* setting. siRNA targeting ERK1/2 was applied and 30 h later, cells were exposed to phenylephrine as an hypertrophic stimulus which induced a hypertrophic response in these cells that were depleted of ERK1/2 to a significant degree. We added an additional sentence and citation to clarify this experiment (**page 11**).

28) The authors use PE as a typical activator of ERK(T188), however the degree of T188 phosphorylation as shown in Figure 4C is quite modest. A summary panel indicating fold-activation across multiple experiments would be helpful. That panel could also include a quantitative indication of EDI's effect on T188 activation, which is difficult to discern in the current immunoblot.

We now refer to the quantification of the immunoblots in the Supplementary Material (**Suppl. Fig. 6B**) and included a different representative immunoblot to better visualize the average signals.

29) Is the disproportional phosphorylation of the T188 site (vs. the TEY motif site) characteristic of tumor types other than colorectal cancer? Replication of some findings in a second tumor type in which MEK-ERK activation is oncogenic (e.g. melanoma) would be even more convincing (though are not absolutely necessary).

This is an intriguing question. Indeed, we have observed similarly disproportional ERK phosphorylation patterns (TEY vs T188) in another tumor. We have now included Western blots of lung tumor and control samples as well as a representative immunohistochemical staining of non-small lung cancer tissue (**Fig. 5A and 5B and Suppl. Fig. 8C**). These experiments validate the occurrence of pERK^{T188} in another type in which MEK-ERK activation is oncogenic, and, a strong upregulation of pERK^{T188} but not pERK(TEY). We believe that pERK^{T188} may be particularly suitable as a biomarker since preliminary experiments suggest that it seems protected from dephosphorylation in the nucleus and may thus be particularly stable.

30) Many of us are accustomed to seeing ERK(TEY) identified by the associated phosphorylation sites (ERK T202/Y204). Could the authors please add one phrase/sentence in which the equivalence is identified?

Thank you for this helpful comment to avoid confusion of potential readers. We have now included a phrase in the

text (**page 4**) and refer to these phosphorylation sites as “TEY”-motif that applies to both ERK isoforms and species.

31) The authors clearly have given careful thought to the title and needn't change it, but I am not sure that their data fully support the inclusion of “at the nucleocytoplasmic interface”.

We have carefully thought about this suggestion. Our previous and additional data clearly show that EDI engages in a novel mechanistic mode of action compared to the class of newer generation MEK/ERK inhibitors (**Fig. 5, Suppl. Fig. 9** and **R2**), which manifests in clear differences in effects on nuclear ERK localization, and nuclear versus cytosolic ERK target and gene expression. If this Reviewer concurs, we therefore feel that is justified to state in the title that EDI constitutes a novel strategy of controlling ERK functions at the nucleocytoplasmic interface.

REVIEWERS' COMMENTS:

Reviewer #1 (Remarks to the Author):

Revised manuscript is much improved

minor comments:

- In the intro text has been changed to "compounds can develop resistances" - I think the authors mean "individuals can develop resistances to these compounds"

- page 18 paragraph 2, "unchanged in lung tumor" - tumor should be pluralized

-Legend for figure 2D should read "western blots were"

Reviewer #2 (Remarks to the Author):

Accept manuscript as presented

Reviewer #3 (Remarks to the Author):

The revised manuscript has addressed all the concerns. The study identified a novel means to inhibit pathological roles of ERK in the nucleus without affecting physiological roles in the cytosol and as such represents advances in the basic understanding of ERK functions and provides a potential lead for drug development.

Reviewer #4 (Remarks to the Author):

The authors have addressed my comments admirably and I have no further concerns. They are to be congratulated for their strong contribution to this field.

Point by point response to reviewers' comments

We are grateful for the positive comments by all Reviewers and their interest in our work. We have included all issues of the remaining comments in the manuscript.

Changes in the manuscript text are displayed using the "tracked changes" feature of Word.

**Reviewer #1:
Revised manuscript is much improved.**

Thank you for positive response and your valuable input for our study.

Minor comments:

1) In the intro text has been changed to "compounds can develop resistances" - I think the authors mean "individuals can develop resistances to these compounds"

Thank you for this important comment. We corrected the sentence (page 4).

2) page 18 paragraph 2, "unchanged in lung tumor" - tumor should be pluralized

Thank you for the comment. We corrected the sentence (page 19).

3) Legend for figure 2D should read "western blots were"

Thank you for the comment. We corrected the sentence (page 48).

**Reviewer #2:
Accept manuscript as presented**

We are glad that we could to answer all important issues. Thank you for your valuable comments!

Reviewer #3:

The revised manuscript has addressed all the concerns. The study identified a novel means to inhibit pathological roles of ERK in the nucleus without affecting physiological roles in the cytosol and as such represents advances in the basic understanding of ERK functions and provides a potential lead for drug development.

Thank you for positive response and your valuable input for our study.

**Reviewer #4:
The authors have addressed my comments admirably and I have no further concerns. They are to be congratulated for their strong contribution to this field.**

Thank you for your congratulations. Yours and the other reviewers' comments substantially helped to strengthen our study.